# Center of Gravity-Guided Focusing Influence Mechanism for Multi-Agent Reinforcement Learning

## Abstract

Cooperative multi-agent reinforcement learning (MARL) under sparse rewards presents a fundamental challenge due to limited exploration and insufficiently coordinated attention among agents. To address this, we introduce the Focusing Influence Mechanism (FIM), a framework that drives agents to concentrate their influence to solve challenging sparse-reward tasks. FIM first identifies Center of Gravity (CoG) state dimensions, inspired by Clausewitz's military strategy, which are prioritized because when they include task-relevant variables, their low variability can block learning unless agents sustain influence. To encourage persistent and synchronized influence, FIM then focuses agents' attention on these CoG dimensions using eligibility traces that accumulate credit over time. These mechanisms enable agents to induce more targeted and effective state transitions, facilitating robust cooperation even under extremely sparse rewards. Empirical evaluations across diverse MARL benchmarks demonstrate that FIM significantly improves cooperative performance over strong baselines.

## 1 Introduction

Cooperative multi-agent reinforcement learning (MARL) has emerged as a powerful framework for sequential decision-making problems involving multiple agents, with applications in autonomous driving (Shalev-Shwartz et al., 2016), multi-robot coordination (Perrusquía et al., 2021), and real-time strategy games (Vinyals et al., 2019). These environments typically involve partial observability, making decentralized partially observable Markov decision processes (Dec-POMDPs) (Oliehoek et al., 2016) a natural modeling choice. To address the challenges arising from limited observability, the centralized training with decentralized execution (CTDE) (Oliehoek et al., 2008; Yu et al., 2022; Sunehag et al., 2018; Rashid et al., 2018; Wang et al.) paradigm has been widely adopted. In CTDE, policies are trained using access to the global state and all agents' observations, but are executed independently using only local observations. Prominent CTDE methods such as VDN (Sunehag et al., 2018), QMIX (Rashid et al., 2018), and QPLEX (Wang et al.) leverage value decomposition to promote coordinated policy learning.

Despite their success, CTDE-based methods often struggle in sparse reward settings where effective exploration is essential (Jaques et al., 2019; Wang et al., 2020b; Liu et al., 2021). Several approaches have been proposed to address this challenge, including maximizing mutual influence between agents (Wang et al., 2020b), prioritizing under-visited but important states (Zheng et al., 2021), and diversifying trajectory distributions (Li et al., 2021a). While promising, we observe that these methods often fail in challenging environments where the state dimensions that agents must eventually influence for task completion do not exhibit diverse changes under typical behaviors, especially in extremely sparse settings, preventing agents from discovering critical transitions and escaping local optima. Thus, we explicitly target environments where the lack of diversity in key elements makes task completion particularly difficult, for example, tasks that require all agents to focus their efforts on a single object to make progress, or settings where agents fall into local optima and never discover the critical elements needed for task success.

To formalize this perspective, we draw on Clausewitz's military theory (Echevarria, 2003), which introduced the concept of the Center of Gravity (CoG) as the focal point where concentrating efforts is most decisive for strategic success. Inspired by this idea, we propose the Focusing Influence Mechanism (FIM), a framework that enhances cooperation by first identifying CoG state dimensions, which are state dimensions that do not exhibit diverse changes under typical agent behaviors

and are individually hard to alter, and then guiding agents to concentrate their influence on them. These dimensions often include task-related variables that are essential for task completion, and if left uninfluenced, they can prevent agents from making progress. FIM addresses this by explicitly selecting such dimensions and maintaining persistent and synchronized influence using eligibility traces that accumulate credit over time, enabling agents to change these otherwise stagnant elements. Concretely, FIM integrates three components: (i) a state-level focusing mechanism that detects CoG dimensions based on their low sensitivity to individual actions, (ii) counterfactual intrinsic rewards that measure each agent's marginal contribution to influencing these dimensions and align local behaviors with team-level goals, and (iii) an agent-level focusing mechanism that sustains coordinated influence through eligibility traces. Together, these components allow agents to consistently affect critical parts of the environment, induce targeted state transitions, and achieve robust cooperation even under extremely sparse rewards. Extensive experiments across diverse MARL benchmarks demonstrate that FIM achieves more efficient collaborative performance than existing methods.

## 2 RELATED WORKS

**Intrinsic Motivation in Sparse Reward MARL**   Intrinsic motivation is widely used to promote exploration in sparse-reward environments. Curiosity-driven objectives encourage agents to seek novel or uncertain states (Iqbal and Sha, 2019; Zheng et al., 2021; Li et al., 2023; Zhang et al., 2023; Yang et al., 2024; Xu et al., 2024), while trajectory diversity methods aim to expand state-space coverage (Zhang and Yu, 2023; Li and Zhu, 2025b;a). Committed exploration is induced by conditioning agent behavior on a shared latent variable (Mahajan et al., 2019), and spatial formation strategies reduce redundant exploration (Jo et al., 2024). Subgoal-based methods decompose tasks into smaller, manageable objectives (Tang et al., 2018; Jeon et al., 2022). Exploration can also be focused in low-dimensional subspaces (Liu et al., 2021; Xu et al., 2023; He et al., 2024), and expectation alignment allows agents to adapt based on anticipated behaviors of peers (Ma et al., 2022).

**Influence-Driven Coordination**   Influence-based methods aim to promote coordination by inducing causally significant changes. Social influence frameworks quantify how an agent's actions affect the behaviors of its teammates (Jaques et al., 2019; Li et al., 2022; Hou et al., 2025) and guide communication decisions (Ding et al., 2020). Opponent modeling enables agents to influence policy updates of others (Foerster et al., 2018a; Letcher et al., 2019; Xie et al., 2021; Kim et al., 2022). Influence-aware exploration affect future dynamics (Wang et al., 2020b; Liu et al., 2024) or induce novel observations (Jiang et al., 2024). Influence has been extended to incentivize beneficial behaviors in others (Yang et al., 2020), discourage undesirable actions (Schmid et al., 2021), or shape the expected returns of other agents (Zhou et al., 2024), as well as to affect external states (Liu et al., 2023) or latent representations of the environment (Li et al., 2024).

**Counterfactual Reasoning Based Credit Assignment**   Counterfactual reasoning facilitates credit assignment by measuring each agent's contribution to the team's shared reward. COMA estimates individual action advantages using counterfactual baselines (Foerster et al., 2018b; Cohen et al., 2021; Wang et al., 2021a; Hoppe et al., 2024), while predictive counterfactual models support value decomposition (Zhou et al., 2022; Chai et al., 2024). Shapley value–based methods assign local credit by marginalizing individual contributions to the global reward (Wang et al., 2020a; Li et al., 2021b; Wang et al., 2022). In offline settings, counterfactual conservatism (Shao et al., 2023) and sample averaging (Ma and Wu, 2023) improve learning stability. Counterfactual reasoning also aids in identifying important agents (Chen et al., 2025) and salient state (Cheng et al., 2023).

## 3 PRELIMINARY

**Decentralized POMDP and CTDE Setup**   In MARL, the environment is typically modeled as a Dec-POMDP (Oliehoek et al., 2016), defined by the tuple $\langle \mathcal{N}, S, A, P, R, O, \mathcal{O}, \gamma \rangle$, where $\mathcal{N}$ is a set of $n$ agents, $S$ is the global state space, $A = A^0 \times \cdots \times A^{n-1}$ is the joint action space, and $\gamma$ is the discount factor. At each timestep $t$, each agent $i \in \mathcal{N}$ receives a local observation $o_t^i = \mathcal{O}(s_t, i)$ and chooses an action $a_t^i$ from its policy $\pi^i$, based on its trajectory $\tau_t^i = (o_0^i, a_0^i, \ldots, o_t^i)$. The state $s_t$ is defined as a $D$-dimensional vector, i.e., $s_t = (s_t^0, \cdots, s_t^{D-1})$, and for given $(s_t, \mathbf{a}_t)$ pair, the environment transitions to $s_{t+1} \sim P(\cdot \mid s_t, \mathbf{a}_t)$ and returns a shared reward $r_t = R(s_t, \mathbf{a}_t)$. The goal is to learn a joint policy $\boldsymbol{\pi} = \prod_{i=1}^{n} \pi^i$ that maximizes the expected return $\sum_{t=0}^{\infty} r_t$. In this paper, we adopt the CTDE paradigm (Rashid et al., 2018), where agents are trained using global state to optimize a total value function $Q^{\text{tot}}$, while each agent executes actions based solely on local observations during deployment.

**Credit Assignment via Counterfactual Reasoning** In the CTDE paradigm, credit assignment mechanisms (Rashid et al., 2018; Foerster et al., 2018b; Shao et al., 2023; Liu et al., 2023) estimate each agent's contribution to team performance, supporting not only the optimization of a global value function but also promoting effective exploration (Li et al., 2021a), information sharing (Jo et al., 2024), and communication (Wang et al., 2020c). A widely adopted technique is counterfactual reasoning (Foerster et al., 2018b; Shao et al., 2023; Liu et al., 2023), which quantifies causal influence by comparing the actual outcome to a counterfactual one where only an individual agent's action is replaced. COMA (Foerster et al., 2018b), for example, defines credit for agent $i$ as:

$$\text{credit}_t^i = f(s_t, \boldsymbol{\tau}_t, \mathbf{a}_t) - \mathbb{E}_{a_t^i \sim P}\left[f(s_t, \boldsymbol{\tau}_t, a_t^i, \mathbf{a}_t^{-i})\right], \tag{1}$$

where $f = Q^{tot}$ and $P = \pi^i(\cdot|s_t)$. This formulation can generalize to any differentiable objective and has been leveraged not only for advantage estimation but also for shaping exploration and coordination via intrinsic rewards.

**Eligibility Trace** Eligibility traces are used to implement TD($\lambda$) online by propagating the current TD error to future timesteps for value updates (Sutton and Barto, 2018). At each timestep $t$, the trace $e_t(s)$ is updated as:

$$e_t(s) = \begin{cases} \gamma\lambda e_{t-1}(s) + 1, & \text{if } s = s_t, \\ \gamma\lambda e_{t-1}(s), & \text{otherwise,} \end{cases} \tag{2}$$

where $\lambda$ is the decay factor. This mechanism accumulates eligibility for recently visited states and decays it over time, focusing value updates on frequently visited states. In this work, we adapt this concept to promote persistent influence on critical states. By extending eligibility traces, we ensure that states with high influence in earlier steps continue to receive attention in subsequent steps, facilitating sustained coordination on task-relevant states.

## 4 METHODOLOGY

### 4.1 MOTIVATION: THE NEED FOR FOCUSING INFLUENCE IN COOPERATIVE MARL

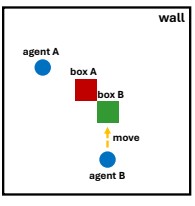 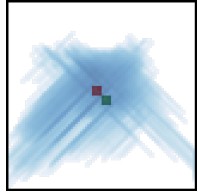 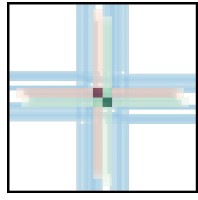 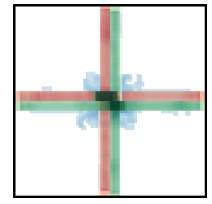

| (a) Push-2-Box | (b) Vanilla QMIX | (c) QMIX with SFI | (d) QMIX with SFI and AFI (proposed FIM) |

Figure 1: Comparative results in the Push-2-Box environment: (a) shows an enlarged view of the environment, and (b–d) show average visitation counts of two agents (blue) and two boxes (Box A: red, Box B: green) over 3M timesteps across 100 seeds. Darker areas indicate more frequent visits.

In cooperative MARL, agents are often required to solve tasks that cannot be accomplished individually, making effective coordination essential (Jaques et al., 2019; Wang et al., 2020b; Liu et al., 2021). Although CTDE algorithms promote cooperation through centralized training, they often fail in sparse reward settings where agents struggle to discover meaningful joint behaviors. To illustrate this challenge, we consider the Push-2-Box environment shown in Fig. 1(a), which involves two agents and two boxes. The task requires both agents to jointly push a single box to the wall within the episode limit to obtain a reward. Because each box moves only one cell when pushed individually and two cells when pushed jointly, coordinated pushing is crucial for success. However, in the absence of intermediate rewards, agents rarely discover the need to push the same box together, leading to almost no variation in the box position dimension during training. Consequently, this task-related state remains nearly static under typical behaviors, making it difficult for agents to explore the transitions necessary for task completion. Fig. 1(b) illustrates this phenomenon, showing scattered exploration and poor coordination, resulting in task failure.

This observation underscores the importance of guiding agents to influence state dimensions that do not exhibit diverse changes under typical behaviors, particularly those that require joint effort to change. To this end, we propose the Focusing Influence Mechanism (FIM), which promotes cooperative behavior through two key components: **state focusing influence (SFI)** and **agent focusing**

**influence (AFI)**. First, SFI identifies Center of Gravity (CoG) state dimensions, which show little diversity under behavior policies to solve challenging tasks that contain task-related variables with limited diversity and are essential for task completion. Inspired by Clausewitz's military theory (Echevarria, 2003), we apply an entropy-based criterion to select these dimensions and guide exploration toward them. We then design a counterfactual intrinsic reward that quantifies each agent's contribution to influencing the CoG dimensions, encouraging alignment of local actions with shared objectives. As shown in Fig. 1(c), incorporating SFI into QMIX allows agents to more frequently influence dimensions such as box positions, which do not exhibit diverse changes unless acted upon cooperatively. However, when multiple CoG dimensions are present, agents tend to alternate their focus, leading to unstable coordination and frequent task failures. To address this, AFI reinforces synchronized and persistent attention to a shared CoG dimension using eligibility traces, stabilizing collective behavior and reducing target switching. Fig. 1(d) shows that QMIX with both SFI and AFI enables agents to maintain focus on a single box and successfully complete the task.

While prior work has explored ways to influence states or coordinate agents (Li et al., 2021a; Wang et al., 2021b; Jeon et al., 2022; Liu et al., 2023; Jo et al., 2024), many rely on heuristics or fail under truly sparse rewards. In contrast, FIM offers a unified framework that combines principled CoG dimension selection, targeted counterfactual intrinsic rewards, and persistent multi-agent attention via eligibility traces. These components enable more purposeful exploration and robust cooperation, and the next section presents each component of FIM in detail.

## 4.2 STATE FOCUSING INFLUENCE VIA CoG STATE DIMENSION SELECTION

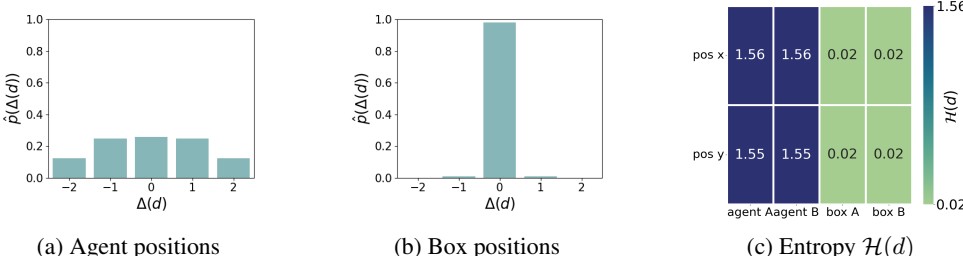

(a) Agent positions  (b) Box positions  (c) Entropy $\mathcal{H}(d)$

Figure 2: (a–b) Empirical distribution $\hat{p}$ of temporal changes $\Delta(d)$ for agent and box positions, averaged over $x, y$ axes. (c) Entropy $\mathcal{H}(d)$ for each of the 8 state dimensions: $(x, y)$ positions of agent A, agent B, box A, and box B in the Push-2-Box environment. We set the threshold $\delta$ to 0.1.

To address the challenge presented in Section 4.1, we focus on tasks such as Push-2-Box that require agents to actively modify state dimensions that are inherently difficult to change. Focusing exploration on such hard-to-influence dimensions is especially beneficial for solving these tasks. From the perspective of value-based RL, it is also well known that good convergence requires visiting a sufficiently diverse set of states (Sutton and Barto, 2018). From an information-theoretic viewpoint, under-explored state dimensions naturally correspond to those with low transition entropy. Given a joint behavior policy $\boldsymbol{\beta}$ and its induced state distribution $\rho^{\boldsymbol{\beta}}$, our goal is therefore to identify dimensions whose next-state variability is small and encourage additional exploration along them, which we formalize via the expected conditional entropy $\mathbb{E}_{s_t \sim \rho^{\boldsymbol{\beta}}, \mathbf{a}_t \sim \boldsymbol{\beta}} \big[ \mathcal{H}(s_{t+1}^d \mid s_t, \mathbf{a}_t) \big]$. For brevity, we denote this expectation by $\mathbb{E}_{\boldsymbol{\beta}}[\cdot]$.

However, directly comparing raw entropies across dimensions is inappropriate, since different dimensions can have different scales of change. For example, letting $U(a, b)$ denote the uniform distribution on $[a, b]$, we have $\mathcal{H}(U(-4, 4)) = \mathcal{H}(U(-2, 2)) + \log 2$ solely because of the larger support, even though from an exploration perspective both are maximally uncertain relative to their typical change. To remove this scale dependence, we compare dimensions using entropy normalized by their average change magnitude $\mathbb{E}_{\boldsymbol{\beta}}[|s_{t+1}^d - s_t^d|]$, so that our criterion is insensitive to scale and captures how under-explored a dimension is relative to how much it tends to move.

**CoG State Dimension Selection.**  To begin, we define the normalized state $\tilde{s}_t = (\tilde{s}_t^0, \ldots, \tilde{s}_t^{D-1})$ with $\tilde{s}_t^d := s_t^d / \mathbb{E}_{\boldsymbol{\beta}}[|s_{t+1}^d - s_t^d|]$. The corresponding dimension-wise entropy of the next state can then be written as $\mathbb{E}_{\boldsymbol{\beta}} \big[ H(\tilde{s}_{t+1}^d \mid \tilde{s}_t, \mathbf{a}_t) \big]$. For simplicity, we omit the explicit dependence on $\boldsymbol{\beta}$ in the notation. Computing this conditional entropy explicitly for all dimensions is still costly, so in practice we approximate it using the normalized state difference and its empirical distribution, as detailed below. We then define the normalized temporal change as

$$\Delta^d(s_t, s_{t+1}) = \tilde{s}_{t+1}^d - \tilde{s}_t^d, \tag{3}$$

and define the entropy of this normalized change as

$$\mathcal{H}(d) = \mathbb{E}_{\boldsymbol{\beta}}\big[-\log \hat{p}\big(\Delta^d(s_t, s_{t+1}) \mid \tilde{s}_t^d\big)\big], \tag{4}$$

where $\hat{p}(\cdot \mid \tilde{s}_t^d)$ is the empirical distribution of $\Delta^d$ conditioned on $\tilde{s}_t^d$, estimated from trajectories under $\boldsymbol{\beta}$. The following theorem shows that this state-difference-based entropy $\mathcal{H}(d)$ serves as a valid surrogate for the theoretically ideal normalized transition entropy.

**Theorem 4.1** *The state-difference-based entropy $\mathcal{H}(d)$ provides an upper bound on the normalized transition entropy under the joint behavior policy $\boldsymbol{\beta}$:*

$$\mathbb{E}_{\boldsymbol{\beta}}\big[\mathcal{H}\big(\tilde{s}_{t+1}^d \mid \tilde{s}_t, \mathbf{a}_t\big)\big] \leq \mathcal{H}(d), \tag{5}$$

*where equality holds when $I\big(\Delta^d(s_{t+1}, s_t); \tilde{s}_t, \mathbf{a}_t \mid \tilde{s}_t^d\big) = 0$.*

**Proof)** Proof is provided in Appendix C.

By Theorem 4.1, we can therefore approximately replace the ideal normalized transition entropy with the state-difference based entropy $\mathcal{H}(d)$ when selecting CoG dimensions, obtaining a tractable yet theoretically justified criterion. Based on these entropy values, we define the CoG set as

$$\mathrm{CoG}_\delta = \{\, d \mid 0 < \mathcal{H}(d) < \delta,\ d = 0, \dots, D-1 \,\}, \tag{6}$$

where $\delta$ is a threshold, and dimensions with zero entropy are excluded as they remain unchanged regardless of agent actions. In this work, the behavior policy $\boldsymbol{\beta}$ is taken from the policy obtained during training, and it can either be kept fixed or updated dynamically. In our main experiments, we compared these two setups in Appendix I.3 and found that, on the standard benchmarks we consider, the difference between fixed and dynamic $\boldsymbol{\beta}$ is negligible, so we adopt the fixed version for simplicity. We also constructed an additional scenario in Appendix I.3 where the influenceable dimensions change over time and showed that in such a case the dynamic setup becomes advantageous, illustrating that our framework can naturally accommodate both fixed and evolving behavior policies.

**SFI Design:** To encourage agents to influence these low-entropy CoG dimensions, we design the following counterfactual intrinsic reward:

$$\mathrm{Inf}_t^d(s_t, \mathbf{a}_t, s_{t+1}) = \sum_{i=0}^{n-1}\Big\{\big|\hat{s}_{t+1}^d(s_t, \mathbf{a}_t) - s_t^d\big| - \mathbb{E}_{a_t^i \sim \beta^i}\big[\big|\hat{s}_{t+1}^d(s_t, a_t^i, \mathbf{a}_t^{-i}) - s_t^d\big|\big]\Big\}, \quad d \in \mathrm{CoG}_\delta, \tag{7}$$

where $\hat{s}(\cdot)$ is a learned dynamics model approximating the transition dynamics $P$, and $\beta^i$ is the behavior policy for agent $i$ used to simulate counterfactual interventions without coordination by agent $i$, as introduced in Section 3. Because low-entropy dimensions are typically characterized by limited change under $\boldsymbol{\beta}$, directly increasing the magnitude of state transitions in these dimensions naturally leads to increased entropy. Thus, even without explicitly maximizing entropy, our reward effectively encourages agents to explore and influence these stable components, which often coincide with important aspects of cooperative tasks. As a result, agents are guided to discover causally meaningful interactions, which improves exploration efficiency and promotes coordinated behavior in sparse-reward environments.

To visualize the proposed SFI described above, we illustrate the process using the Push-2-Box environment. Fig. 2 shows the empirical distribution $\hat{p}$ of state changes $\Delta(d)$ for (a) agent positions, (b) box positions, and (c) the corresponding entropy of each state dimension. As shown in Fig. 2(c), agent positions, being directly controlled, vary frequently and exhibit high entropy, while box positions change only through coordinated effort, resulting in low entropy. Using a threshold of $\delta = 0.1$, the $x$ and $y$ positions of the box are selected as CoG state dimensions. When the sum of proposed intrinsic reward $\sum_{d \in \mathrm{CoG}_\delta} \mathrm{Inf}_t^d$ is applied, agents focus on these dimensions, leading to more frequent and diverse box movement, as illustrated in Fig. 1(c). This example demonstrates how our method identifies hard-to-change dimensions that require joint effort, which in this environment align with task-relevant components. In Section 5 and Appendix F.3, we analyze how CoG state dimensions are selected in complex environments and compare our selection method with naive and prior heuristic approaches to show its effectiveness.

## 4.3 AGENT FOCUSING INFLUENCE BASED ON ELIGIBILITY TRACE

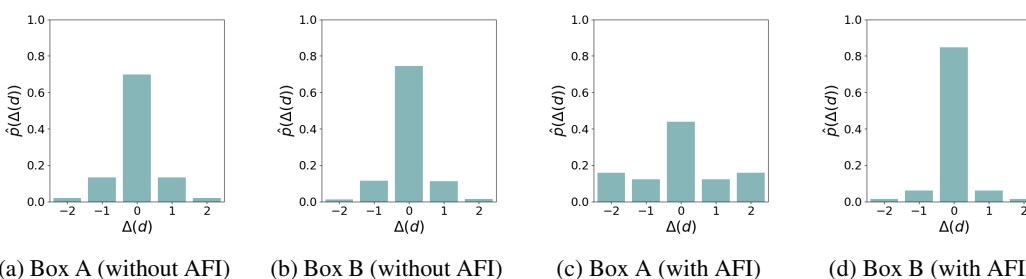

(a) Box A (without AFI)    (b) Box B (without AFI)    (c) Box A (with AFI)    (d) Box B (with AFI)

Figure 3: Empirical distribution $\hat{p}$ of temporal changes $\Delta(d)$ for CoG dimensions (box positions): (a–b): Without AFI. (c–d): With AFI, where Box A is the focused target.

While the proposed SFI guides agents to actively influence CoG state dimensions that show limited changes under the behavior policy, coordination often becomes unstable when multiple such dimensions are present. Agents may alternate attention across them without maintaining focus, leading to scattered and ineffective behavior. This issue is evident in the Push-2-Box environment introduced in Section 4.1, where agents frequently switch between the two boxes and fail to push either to the wall. Such inconsistency is particularly problematic in tasks that require all agents to jointly influence a single object. To address this, we propose agent focusing influence (AFI), a mechanism that promotes persistent and synchronized attention on a shared CoG dimension through eligibility traces. Specifically, we quantify the current influence on each dimension $d$ as $\mathrm{Inf}_t^d$ and update the eligibility trace $e_t^d$ over time as:

$$e_t^d = \lambda \cdot e_{t-1}^d + \eta \cdot \mathrm{Inf}_t^d,\ d \in \mathrm{CoG}_\delta, \tag{8}$$

where $\lambda \in [0,1]$ is a decay factor and $\eta > 0$ is a scaling coefficient. The trace $e_t$ accumulates historical influence until time $t$, increasing as agents repeatedly affect the same dimension.

To guide agents to concentrate on such dimensions, we define an intrinsic reward:

$$r_{\mathrm{int},t} = \sum_{s^d \in \mathrm{CoG}_\delta} w_d \cdot \mathrm{Inf}_t^d \cdot \mathrm{clip}(e_{t-1}^d, 1, c_{\max}), \tag{9}$$

where $w_d = \mathrm{Softmax}(-\mathcal{H}(d))$ prioritizes lower-entropy (harder-to-change) CoG state dimensions, and the clipping operator $\mathrm{clip}(\cdot, 1, c_{\max})$ ensures reward stability ($c_{\max}$ set to 10). This design encourages agents to reinforce influence on dimensions they have consistently affected, fostering collective persistence. If a previously focused dimension becomes unreachable (e.g., the target is destroyed or removed), its influence naturally drops, shifting agent attention to the next most relevant CoG dimension. Through this mechanism, agents learn to sequentially commit to one shared target at a time, leading to more robust coordination.

To illustrate the effect of the proposed AFI, Fig. 3 shows how the empirical distribution of temporal changes in CoG state dimensions (i.e., box positions) evolves with and without AFI in the Push-2-Box environment. Without AFI (i.e., $\eta = 0$, $w_d = 1$), applying only SFI, (a) and (b) display greater variation in both boxes compared to vanilla QMIX in Fig. 2(b), indicating increased interaction with CoG dimensions. However, due to lack of focus on a single box, agents split their influence, leading to unstable coordination and task failure. With both SFI and AFI, (c) and (d) show that agents collectively concentrate on Box A, resulting in significantly more variation in its position, while Box B remains mostly unchanged. This focused influence increases entropy for Box A, aligning with successful task completion in Fig. 1(d). This mechanism enables agents to succeed not only in toy tasks but also in more complex multi-agent scenarios. For instance, in combat-style environments, agents can collectively focus on disabling a key opponent, while in soccer-like domains, they may coordinate interference against a specific defender. Even under sparse rewards, this influence-driven reward promotes persistent cooperation and reliable task completion.

By combining the proposed SFI and AFI, we introduce the Focusing Influence Mechanism (FIM) for MARL, which directs each agent's influence toward CoG state dimensions and encourages collective focus on a single target. Agents receive an intrinsic reward $r_{\mathrm{int},t}$ alongside the environment-provided external reward $r_{\mathrm{ext},t}$, forming a total reward $r_t = r_{\mathrm{ext},t} + \alpha r_{\mathrm{int},t}$, where $\alpha$ balances the two terms.

We adopt QMIX (Rashid et al., 2018) as the base learner, though our intrinsic reward is model-agnostic and applicable to other MARL algorithms. Further implementation details and the full algorithm of FIM are provided in Appendix D.2.

## 5 EXPERIMENT

In this section, we evaluate the effectiveness of the proposed FIM. We begin with the Push-2-Box task introduced in Section 4.1, comparing various combinations of our proposed components. We then extend the evaluation to more complex MARL benchmarks, the StarCraft Multi-Agent Challenge (SMAC) (Samvelyan et al., 2019) and Google Research Football (GRF) (Kurach et al., 2020). In all performance plots, the mean across 5 random seeds is shown as a solid line, and the standard deviation is represented by a shaded area. As shown in Appendix. I.3, using the fixed CoG set computed from the initial behavior policy already yields strong performance, so we adopt the fixed CoG configuration in all main experiments.

### 5.1 PERFORMANCE COMPARISON: COMPONENT EVALUATION ON PUSH-2-BOX

We revisit the Push-2-Box task, where two agents must jointly push one of two boxes to a wall, as shown in Fig. 1(a). A box moves by one grid cell if pushed by a single agent and two cells if pushed by both agents. A external reward of +100 is given when either box reaches the wall and -1 is applied if the task fails. The environment is considered successfully solved when agents manage to push a box to the wall within the episode length through synchronized cooperation. More detailed environment settings are provided in Appendix E. Fig. 4 shows the success rate comparison between several baselines: **LAIES** (Liu et al., 2023), which encourages influence over heuristic external state features (i.e. box positions); **CDS** (Li et al., 2021a), which promotes trajectory diversity for exploration; **FoX** (Jo et al., 2024), which leverages formation-aware exploration; vanilla QMIX trained with only extrinsic rewards; QMIX with SFI; QMIX with AFI; and FIM.

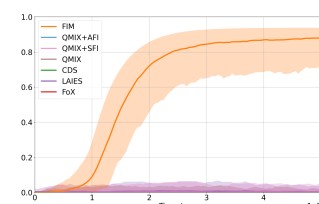

Figure 4: Performance comparison across the proposed focusing components on Push-2-Box environment.

In SFI, agents are guided to influence selected CoG state dimensions using an intrinsic reward $\sum_{d \in \text{CoG}\delta} \text{Inf}_t^d$ that promotes interaction with low-entropy components. In contrast, AFI applies agent-level focusing across all state dimensions without CoG selection, where the intrinsic reward is given by $\sum_{d=0}^{D-1} \text{Inf}_t^d \cdot \text{clip}(e_{t-1}^d, 1, c_{\max})$. FIM combines both selective targeting and synchronized persistence via the intrinsic reward structure in Eq. 9. We observe that only FIM consistently succeeds in solving the task. Vanilla QMIX alone fails due to ineffective exploration. SFI enhances interaction with hard-to-change states requiring joint effort, as illustrated in Fig. 1(c), but struggles to maintain consistent focus on a single target, as seen in Fig. 3, which leads to task failure. AFI promotes sustained influence when combined with SFI, yet fails on its own due to the absence of targeted attention. These results emphasize that both principled state selection and agent-level coordination are essential for effective cooperation in sparse-rewarded environments.

### 5.2 PERFORMANCE COMPARISON ON COMPLEX MARL BENCHMARKS: SMAC AND GRF

Next, we evaluate our method on two complex MARL benchmarks: SMAC and GRF. SMAC is a multi-agent combat environment built on StarCraft II, where agents must coordinate to defeat enemy units. We use a truly sparse reward setting in which agents receive +1 for a win, 0 for a draw, and -1 for a loss. Evaluation is conducted on 8 challenging scenarios: 3 hard maps (`5m_vs_6m`, `8m_vs_9m`, `3s_vs_5z`) and 5 super hard maps (`corridor`, `MMM2`, `6h_vs_8z`, `27m_vs_30m`, `3s5z_vs_3s6z`), where m, s, z, and h refer to marine, stalker, zealot, and hydralisk units, respectively. GRF is a multi-agent soccer environment where teams compete to score goals under sparse rewards: +100 for a win and -1 for a loss. We evaluate on 8 scenarios, including 4 half-field settings (`academy_2_vs_2`, `academy_3_vs_2`, `academy_4_vs_3`, `academy_counterattack`) and their corresponding full-field versions, which are more challenging due to the increased field size. Further environment details and visualizations are provided in Appendix E.

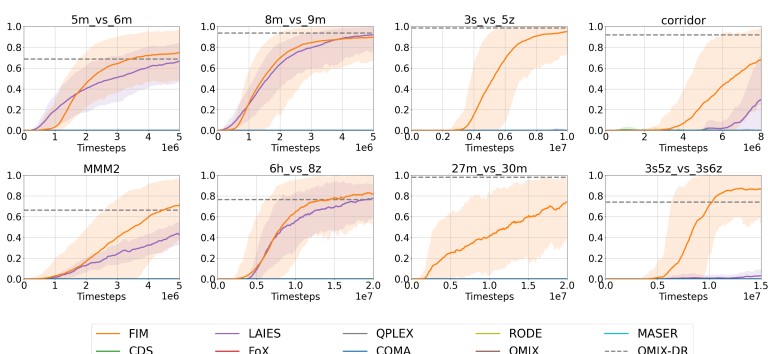

Figure 5: Performance comparison on SMAC environments

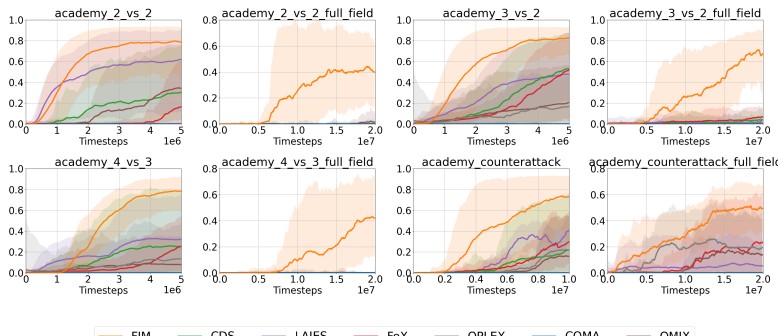

Figure 6: Performance comparison on GRF environments

For SMAC, we compare FIM against several QMIX-based baselines: **Vanilla QMIX** (Rashid et al., 2018); **LAIES** (Liu et al., 2023); **CDS** (Li et al., 2021a); **FoX** (Jo et al., 2024); **COMA** (Foerster et al., 2018b), which uses counterfactual baselines for centralized multi-agent policy gradient; **MASER** (Jeon et al., 2022), which identifies subgoals based on $Q$-values; **RODE** (Wang et al., 2021b), which assigns latent roles to agents; and **QPLEX** (Wang et al.), which applies monotonic value decomposition with a dueling architecture. For GRF, we compare against Vanilla QMIX, LAIES, CDS, FoX, COMA and QPLEX, omitting baselines without publicly available GRF results. We also include QMIX-DR for SMAC, trained under dense reward settings, to provide an upper-bound reference. Baseline algorithms are evaluated using author-provided implementations, while our method uses the best hyperparameter settings identified through ablation studies. Detailed descriptions of each algorithm, our hyperparameter configurations, and CoG state dimensions for each environment are provided in Appendix F.

Fig. 5 and Fig. 6 present success rate comparisons on the SMAC and GRF benchmarks. In SMAC, QMIX-DR is shown only as a final point to indicate an upper bound under dense rewards. Across both environments, FIM consistently achieves the highest success rates among all baselines. In SMAC, the sparse reward setting poses a significant challenge, as agents must eliminate all enemies without intermediate feedback. While LAIES remains competitive in some scenarios, it struggles on complex maps like `27m_vs_30m` and `corridor`, where it prioritizes influence over external states except ally agents. In contrast, FIM demonstrates robustness by selectively targeting dimensions that serve as strategic coordination points. In GRF, although some baselines perform well on simpler half-field tasks, they largely fail on full-field maps with rare scoring opportunities. FIM, by focusing influence on hard-to-change elements, maintains strong performance across all scenarios. These results highlight that FIM promotes effective cooperation, enabling agents to solve challenging tasks even under highly sparse rewards. For practical comparison, we also evaluate the computational complexity of our method against QMIX in Appendix G. The additional result demonstrates that, with nearly the same training time as QMIX, our method achieves superior performance that is unattainable by the baselines. Furthermore, Appendix I.1 presents additional experiments showing that FIM achieves strong performance even in the more challenging SMACv2 and in MPE, where state dimensions are highly dynamic. In both cases, FIM consistently selects relatively stable state dimensions, demonstrating its generality across diverse MARL settings.

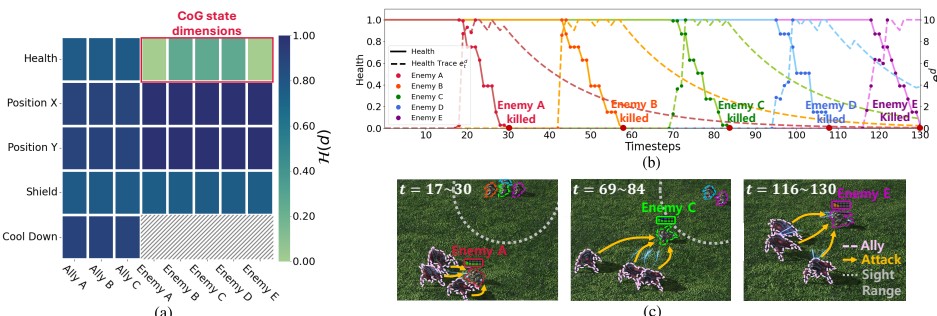

Figure 7: Trajectory analysis in `3s_vs_5z`: (a) Entropy $\mathcal{H}(d)$ with selected CoG state dimensions (b) Changes in enemy health and its trace $e_t^d$ (c) Rendered frames for highlighting agents' coordination.

## 5.3 IN-DEPTH ANALYSIS AND ABLATION STUDIES ON SMAC AND GRF

To better understand the impact of FIM's focusing mechanisms, we conduct detailed analyses and ablations in environments where it shows the largest advantage: SMAC's `3s_vs_5z` and GRF's `academy_3_vs_2_full_field`. In SMAC `3s_vs_5z`, the state focusing mechanism highlights enemy features such as health and shield as CoG state dimensions, since they are relatively stable without coordination, making them natural targets for joint influence. As detailed in Appendix F.3, similar CoG state dimension patterns are observed in other SMAC environments, while in GRF, the keeper's position is frequently selected as a CoG state dimension, as it is challenging for agents to manipulate. These results demonstrate that the proposed method identifies and influences key CoG state dimensions, enhancing performance by focusing on impactful features like health and shield in SMAC and the keeper's position in GRF.

To further illustrate the effect of the proposed method, Fig. 7(a) presents entropy $\mathcal{H}(d)$ values for each dimension in `3s_vs_5z`, where the health of the five enemy units is selected as CoG dimensions with $\delta = 0.1$. These features change significantly only when agents coordinate attacks. Fig. 7(b) shows how eligibility traces evolve on enemy health dimensions during an episode, and Fig. 7(c) visualizes key timesteps where enemy units are eliminated. Agents trained with FIM learn to pull enemies into sight and focus fire sequentially. Around $t \approx 20$, they concentrate on the first red enemy, increasing its health trace, and eliminate it by $t \approx 30$. Once removed, its influence drops to zero, and focus shifts to the next enemy (e.g., orange at $t \approx 40$), repeating this process. This strategy resembles human gameplay in StarCraft II. We also provide analysis for GRF in Appendix H, where the results show that agents learn to disrupt the behavior of the keeper, selected as a CoG state dimension, thereby increasing goal-scoring opportunities. Although we report results with a fixed CoG state dimensions, the entropy-based selection rule can be re-applied during training, which allows the framework to update CoG state dimensions when relevance shifts. These findings demonstrate how FIM promotes structured and effective cooperation even in sparse-reward environments.

Beyond visualization, we conduct ablation studies on `3s_vs_5z` to evaluate the contributions of each component and the sensitivity to key hyperparameters. Fig. 8(a) compares performance across the variants considered in Fig. 4: Vanilla QMIX, QMIX with SFI, QMIX with AFI, LAIES+SFI, LAIES+AFI and the full FIM combining both. LAIES+SFI replaces LAIES's extrinsic state with our CoG-selected dimensions and LAIES+AFI rescales LAIES's influence using per-dimension eligibility traces accumulated over time. Results also show that while SFI and AFI individually improve performance, combining them leads to faster convergence and higher final success rates, confirming the synergy between selective state targeting and synchronized agent coordination. Fig. 8(b)–(d) further examine the effects of the trace scaling factor $\eta$, intrinsic reward weight $\alpha$, and trace decay factor $\lambda$. Performance is sensitive to these parameters: too-small values weaken intrinsic rewards and hinder learning, while overly large values lead agents to overfit intrinsic signals and ignore extrinsic rewards. This trade-off is common in intrinsic-motivation-based methods, emphasizing the importance of proper scaling. We set $\eta = 50$, $\alpha = 1$, and $\lambda = 0.95$ as default values based on observed performance. To further evaluate the effectiveness of the proposed method, we provide ablation studies comparing FIM with naive and heuristic state selection approaches, along with results from other environments in Appendix I.2. We also include in Appendix I.4 an analysis of applying SFI and AFI components to LAIES, further clarifying their individual contributions to influence-

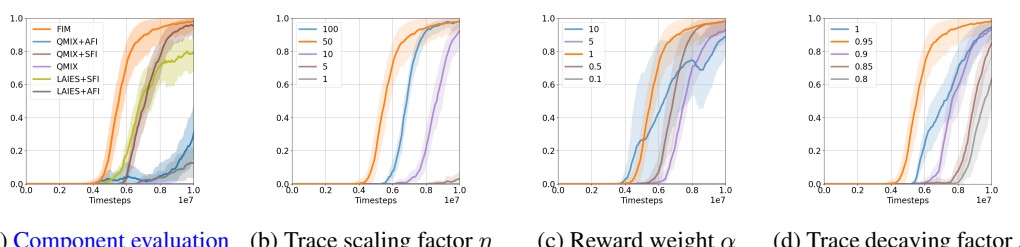

(a) Component evaluation    (b) Trace scaling factor $\eta$    (c) Reward weight $\alpha$    (d) Trace decaying factor $\lambda$

Figure 8: Ablation studies on SMAC `3s_vs_5z`

based exploration. We also provide in Appendix J an analysis of the learned dynamics model used by FIM, showing how prediction error evolves during training and how its behavior relates to exploratory effectiveness and overall performance. FIM consistently outperforms all baselines, further demonstrating the superiority of the CoG state dimension selection method and the overall FIM framework.

# 6 LIMITATION

Although FIM achieves strong performance, it shares some common limitations of intrinsic-motivation–based methods. First, performance can be sensitive to hyperparameter choices such as the intrinsic reward weight $\alpha$, trace decay factor $\lambda$, and scaling coefficient $\eta$. While we provide ablation studies and default settings, additional tuning may be required in new domains. Second, although the additional computational cost of training the dynamics model is modest compared to QMIX (see Appendix G), it still introduces overhead in large-scale applications. Addressing these issues through more robust hyperparameter adaptation and lightweight model approximations would further improve practicality.

# 7 CONCLUSION

In this paper, we address the challenge of efficient cooperation in sparse-reward MARL by proposing FIM, a framework that guides agent influence toward CoG state dimensions and sustains coordinated focus through eligibility traces. By integrating principled state selection with structured intrinsic rewards based on counterfactual reasoning, FIM enables agents to induce targeted and persistent state transitions. Empirical results across Push-2-Box, SMAC, and GRF demonstrate that FIM significantly improves learning efficiency and coordination, outperforming state-of-the-art baselines. These findings highlight the potential of influence-guided learning to enable robust multi-agent cooperation in complex and sparsely rewarded environments.

## ETHICS STATEMENT

This paper introduces the Focusing Influence Mechanism (FIM) for cooperative multi-agent reinforcement learning, evaluated entirely in simulated benchmark environments (Push-2-Box, SMAC/SMACv2, GRF, and MPE). The work does not involve human subjects, personally identifiable or sensitive data, or applications that directly interact with people. As such, issues of privacy, discrimination, or fairness are not directly applicable. We also confirm that our experiments comply with legal, research integrity, and ethical standards. We note that while our research poses no immediate risks.

## REPRODUCIBILITY STATEMENT

We are committed to ensure reproducibility of our results. The complete source code for FIM, including training scripts, environment wrappers, and configuration files, is provided in the anonymized supplementary materials. Algorithmic details are presented in Section 4 and Appendix D.1, with the full procedure summarized in Algorithm 1. Environment specifications are given in Appendix E, hyperparameters and baseline configurations in Appendix F, and hardware/software settings in Appendix G. Additional ablation studies and generalization results are provided in Appendix I.2 and Appendix I.1. These resources together provide all necessary information for independent reproduction and verification of our findings.

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

## A    THE USE OF LARGE LANGUAGE MODELS

In preparing this manuscript, we used a large language model (LLM) solely as an assistive tool for polishing the final text. Specifically, the LLM was employed to improve grammar, style, and clarity of exposition. It was not used for research ideation, experimental design, theoretical development, or analysis of results. All scientific content, algorithms, and experiments were conceived, implemented, and validated entirely by the authors. The authors have thoroughly reviewed and edited all text, and take full responsibility for the content of this manuscript. The LLM is not credited as an author.

## B    BROADER IMPACT

This work advances cooperative multi-agent systems by introducing a framework that fosters coordinated behavior through influence-based intrinsic motivation. Enhanced cooperation among agents holds strong potential for positive societal impact in domains such as autonomous vehicle coordination, collaborative robotics, disaster response, and environmental monitoring. In these settings, the ability of agents to reason about and influence task-critical aspects collaboratively can lead to more robust, adaptive, and efficient team performance. As a foundational contribution, this research supports the development of AI systems that are better aligned with collective goals, promoting safer and more effective deployment in real-world multi-agent environments.

## C    PROOF OF THEOREM 4.1

The left-hand side of Eq. 5 is

$$\mathbb{E}_{\boldsymbol{\beta}}\big[\mathcal{H}(\tilde{s}_{t+1}^d \mid \tilde{s}_t, \mathbf{a}_t)\big] = \mathbb{E}_{\boldsymbol{\beta}}\big[\mathcal{H}(\tilde{s}_{t+1}^d - \tilde{s}_t^d \mid \tilde{s}_t, \mathbf{a}_t)\big]$$
$$= \mathbb{E}_{\boldsymbol{\beta}}\big[\mathcal{H}\big(\Delta^d(s_t, s_{t+1})|\tilde{s}_t, \mathbf{a}_t\big)\big]$$
$$= \mathcal{H}\big(\Delta^d(s_t, s_{t+1})|\tilde{s}_t, \mathbf{a}_t\big)$$

The inequality in Eq. 5 follows from the nonnegativity of mutual information $I\big(\Delta^d(s_{t+1}, s_t); \mathbf{a}_t \mid \tilde{s}_t\big)$:

$$I\big(\Delta^d(s_{t+1}, s_t); \mathbf{a}_t \mid \tilde{s}_t\big) = \mathcal{H}\big(\Delta^d(s_{t+1}, s_t) \mid \tilde{s}_t\big) - \mathcal{H}\big(\Delta^d(s_{t+1}, s_t) \mid \tilde{s}_t, \mathbf{a}_t\big) \geq 0,$$

which implies

$$\mathcal{H}\big(\Delta^d(s_{t+1}, s_t) \mid \tilde{s}_t, \mathbf{a}_t\big) \leq \mathcal{H}\big(\Delta^d(s_{t+1}, s_t) \mid \tilde{s}_t\big)$$
$$\leq \mathcal{H}\big(\Delta^d(s_{t+1}, s_t) \mid \tilde{s}_t^d\big)$$
$$= \mathcal{H}(d).$$

Equality holds if and only if the mutual information vanishes, i.e.,

$$I\big(\Delta^d(s_{t+1}, s_t); \tilde{s}_t, \mathbf{a}_t \mid \tilde{s}_t^d\big) = 0,$$

## D    IMPLEMENTATION DETAILS

In this section, we provide practical details on how the proposed framework is implemented. First, we describe the empirical implementation of state focusing influence, including how entropy is estimated, counterfactual influences are approximated, and the transition model is trained in Appendix D.1. Next, we present the overall learning procedure summarized in Appendix D.2.

### D.1    EMPIRICAL IMPLEMENTATION OF STATE FOCUSING INFLUENCE

To estimate $\mathcal{H}(d) = \mathbb{E}_{\boldsymbol{\beta}}\big[-\log \hat{p}\big(\Delta^d(s_t, s_{t+1}) \mid \tilde{s}_t^d\big)\big]$, we approximate the marginal distribution of normalized changes $p\big(\Delta^d(s_t, s_{t+1})\big)$, since conditioning on full states $p\big(\Delta^d(s_t, s_{t+1}) \mid \tilde{s}_t^d\big)$ is

computationally prohibitive. We construct the empirical distribution $\hat{p}\left(\Delta^d(s_t, s_{t+1})\right)$ by counting occurrences discretized to two decimal places from 100K episodes collected under the initial behavior policy. The entropy is then computed as:

$$\mathcal{H}(d) \approx \mathbb{E}_\beta\left[-\log\hat{p}(\Delta^d(s_t, s_{t+1}))\right] \tag{10}$$

To ensure comparability across environments, $\mathcal{H}(d)$ values for $d \in \mathrm{CoG}_\delta$ are min-max normalized to the range $[0, 1]$ within each environment. To assess the validity of using marginal entropy as a surrogate, we empirically measured both the conditional and marginal entropies for each state dimension in GRF `academy_2_vs_2`, as shown in Table 1. We found that the two quantities are numerically very close and, more importantly, induce almost identical rankings across dimensions. Since CoG selection depends only on this ranking, the marginal entropy provides a reliable and statistically efficient surrogate in practice.

| State Dimension | $\mathcal{H}\left(\Delta^d(s_t, s_{t+1})\right)$ | $\mathcal{H}\left(\Delta^d(s_t, s_{t+1}) \mid \tilde{s}_t^d\right)$ |
| --- | --- | --- |
| Ally A position | 0.73 | 0.72 |
| Ally A direction | 0.82 | 0.80 |
| Ally B position | 0.73 | 0.71 |
| Ally B direction | 0.78 | 0.73 |
| Opponent GK position | 0.27 | 0.21 |
| Opponent GK direction | 0.32 | 0.26 |
| Opponent A position | 0.74 | 0.70 |
| Opponent A direction | 0.96 | 0.94 |
| Ball position | 0.47 | 0.45 |
| Ball direction | 0.24 | 0.23 |

Table 1: Comparison of $\mathcal{H}\left(\Delta^d(s_t, s_{t+1})\right)$ and $\mathcal{H}\left(\Delta^d(s_t, s_{t+1}) \mid \tilde{s}_t^d\right)$, averaged over state dimensions, measured under the initial behavior policy in GRF `academy_2_vs_2`.

The counterfactual intrinsic reward in Eq. 7 is computed as the sum of $\mathrm{Inf}_t^{d,i}(\cdot)$ over agents $i$, where $\mathrm{Inf}_t^{d,i}(\cdot)$ measures the influence of agent $i$ on state dimension $s^d$ at time $t$:

$$\mathrm{Inf}_t^{d,i}(s_t, \mathbf{a}_t, s_{t+1}) = \left|\hat{s}_{t+1}^d(s_t, \mathbf{a}_t) - s_t^d\right| - \mathbb{E}_{a_t^i \sim \beta^i}\left[\left|\hat{s}_{t+1}^d(s_t, a_t^i, \mathbf{a}_t^{-i}) - s_t^d\right|\right] \tag{11}$$

The transition model $\hat{s}$ used to compute $\mathrm{Inf}_t^{d,i}(\cdot)$ is implemented as a three-layer multilayer perceptron (MLP) and trained by minimizing the following mean squared error loss:

$$\mathcal{L}_{\hat{s}} = \mathbb{E}_{s_t, \mathbf{a}_t, s_{t+1}}\left[\|\hat{s}(s_t, \mathbf{a}_t) - s_{t+1}\|^2\right] \tag{12}$$

Since the influence is estimated using a learned model, approximation noise can introduce spurious nonzero signals even when agent $i$ has no actual effect on $s^d$. To mitigate false positives, we discard any $\mathrm{Inf}_t^{d,i}(\cdot)$ below a threshold $\kappa$, and mask out agents that are inactive or dead at time $t$. The final influence on dimension $s^d$ is computed by summing only the valid contributions:

$$\mathrm{Inf}_t^d(s_t, \mathbf{a}_t, s_{t+1}) = \sum_{i \in \mathcal{N}_t} \mathbf{1}[\mathrm{Inf}_t^{d,i}(s_t, \mathbf{a}_t, s_{t+1}) \geq \kappa] \cdot \mathrm{Inf}_t^{d,i}(s_t, \mathbf{a}_t, s_{t+1}) \tag{13}$$

where $\mathcal{N}_t$ denotes the set of active agents at time $t$, and $\mathbf{1}[\cdot]$ is the indicator function.

### D.2 COMPLETE IMPLEMENTATION AND ALGORITHMIC DETAILS OF FIM

The FIM framework builds on the centralized training with decentralized execution (CTDE) paradigm, using QMIX to learn a joint action-value function. Each agent maintains an individual $Q$-function $Q^i(\tau_t^i, a_t^i)$ based on its action-observation history $\tau_t^i$ and current action $a_t^i$. These per-agent utilities are combined via a mixing network to produce a global joint $Q$-value, $Q_\theta^{\mathrm{tot}}(s_t, \mathbf{a}_t)$, where $\theta$ denotes the parameters of the mixing network.

To stabilize learning, FIM employs a target mixing network $Q_{\theta^-}^{\text{tot}}$, which is periodically updated by overwriting its parameters with those of the current mixing network. The temporal difference (TD) loss is computed using a Bellman update that incorporates both extrinsic and intrinsic rewards:

$$\mathcal{L}_{\text{TD}}(\theta) = \mathbb{E}_{s,\mathbf{a},r,s'}\left[\left(r_{\text{ext},t} + \alpha r_{\text{int},t} + \gamma \max_{\mathbf{a}'} Q_{\theta^-}^{\text{tot}}(s_{t+1}, \mathbf{a}') - Q_{\theta}^{\text{tot}}(s_t, \mathbf{a}_t)\right)^2\right] \quad (14)$$

This loss is minimized using the Adam optimizer to update the parameters $\theta$, while the target network parameters $\theta^-$ are synchronized at fixed intervals. The complete training procedure of FIM is summarized in Algorithm 1.

---

**Algorithm 1:** FIM framework

---

**Initialize:** Q network, dynamics model $\hat{s}$

1  Collect trajectories under behavior policy
2  Approximate $\mathcal{H}(d)$ with the obtained trajectories based on Eq. (4)
3  Define CoG state dimensions $\text{CoG}_\delta$ based on Eq. (5)
4  Compute $w_d$ for each $d \in \text{CoG}_\delta$
5  **for** training iteration **do**
6      **for** timestep $t$ **do**
7          Sample transition $(s_t, \mathbf{o}_t, \mathbf{a}_t, s_{t+1}, \mathbf{o}_{t+1})$ using $\boldsymbol{\pi}$, where $\mathbf{o}_t = (o_t^0, \cdots, o_t^{n-1})$
8          **for** $d \in \text{CoG}_\delta$ **do**
9              Compute collective influence $\text{Inf}_t^d$ by Eq. (6)
10             Update eligibility trace $e_t^d$ by Eq. (7)
11         Compute intrinsic reward $r_{\text{int},t}$ by Eq. (8)
12     Update value function $Q^{\text{tot}}$ and dynamics model $\hat{s}$

---

# E  ENVIRONMENT DETAILS

## Push-2-Box

Push-2-Box is a cooperative multi-agent environment where two agents must jointly push one of two boxes toward a wall to obtain a reward. A box moves one cell if pushed by a single agent and two cells if pushed simultaneously. Thus, synchronized cooperation is essential for completing the task within the episode time limit. The environment terminates either when a box reaches the wall or when the episode length is exceeded.

The **state space** consists of the $(x, y)$ positions of both agents and both boxes, resulting in an 8-dimensional state vector. To isolate coordination from partial observability, each agent receives the full environment state as observation. The **action space** is discrete, consisting of eight movement actions corresponding to up, down, left, right, top-right, right-down, down-left, and left-top directions. The **reward function** is sparse, providing +100 if a box reaches the wall and -1 if no box reaches the wall by the end of the episode.

**StarCraft Multi-Agent Challenge (SMAC)**

SMAC (Samvelyan et al., 2019) is a benchmark for evaluating decentralized cooperative multi-agent reinforcement learning. Agents control individual StarCraft II units and must coordinate to defeat enemy forces operated by a scripted AI. During centralized training, a global state is accessible, but during decentralized execution, each agent relies solely on its local observations within a limited sight range. SMAC offers both dense and sparse reward settings, with the sparse reward setting significantly increasing the difficulty by removing intermediate feedback.

The **state space** aggregates absolute features of all units, including positions, health, shields, energies, cooldowns, unit types, and past actions. The **observation space** provides each agent with relative $(x, y)$ positions, health, shield status, and unit types of nearby allies and enemies. The **action space** is discrete, consisting of movement in four directions, attacking visible enemies, stopping, and no-op actions (only allowed for dead units). The **reward function** is summarized in Table 2, and our experiments focus on the sparse reward setting across eight challenging scenarios. Scenario visualizations are provided in Fig. 9, with unit compositions and environment dimensions summarized in Table 3.

| Event | Dense reward | Sparse reward |
|---|---|---|
| Enemy unit killed | +10 per enemy killed | No reward |
| Ally unit killed | -10 per ally killed | No reward |
| Damage dealt to enemy | + (proportional to damage amount) | No reward |
| Damage received by ally | - (proportional to damage amount) | No reward |
| Winning the battle | +200 at episode end | +1 at episode end |
| Losing the battle | 0 | -1 at episode end |

Table 2: Comparison of dense and sparse reward structures in SMAC

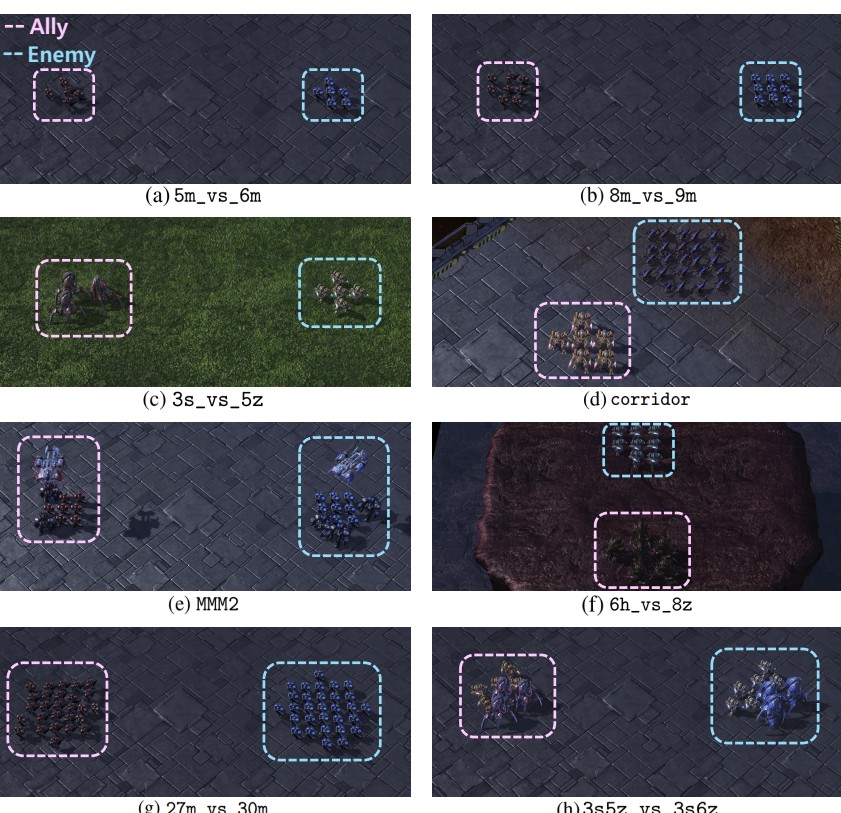

(a) 5m_vs_6m

(b) 8m_vs_9m

(c) 3s_vs_5z

(d) corridor

(e) MMM2

(f) 6h_vs_8z

(g) 27m_vs_30m

(h) 3s5z_vs_3s6z

Figure 9: Visualization of initial timestep in SMAC scenarios.

| Scenario | Ally | Enemy | State Dim | Obs Dim | Action Dim |
|---|---|---|---|---|---|
| 5m_vs_6m | 5 Marines | 6 Marines | 98 | 55 | 12 |
| 8m_vs_9m | 8 Marines | 9 Marines | 179 | 85 | 15 |
| 3s_vs_5z | 3 Stalkers | 5 Zealots | 68 | 48 | 11 |
| corridor | 6 Zealots | 24 Zerglings | 282 | 156 | 30 |
| MMM2 | 1 Medivac, 2 Marauders, 7 Marines | 1 Medivac, 3 Marauders, 8 Marines | 322 | 176 | 18 |
| 6h_vs_8z | 6 Hydralisks | 8 Zealots | 140 | 78 | 14 |
| 27m_vs_30m | 27 Marines | 30 Marines | 1170 | 285 | 36 |
| 3s5z_vs_3s6z | 3 Stalkers, 5 Zealots | 3 Stalkers, 6 Zealots | 230 | 136 | 15 |

Table 3: SMAC scenario configuration

**Google Research Football (GRF)**

GRF (Kurach et al., 2020) provides a realistic soccer simulation, incorporating ball dynamics, passing, shooting, tackling, and player movement mechanics. Agents control individual players on a team and must cooperate to score goals against an opponent team controlled by a scripted AI. We adopt a sparse reward setting to evaluate cooperative behavior under severely limited feedback.

The **state space** during centralized training consists of player positions and velocities, as well as ball position and velocity. Each **observation space** for an agent includes local information about the ego player, nearby teammates, opponents, and ball-related features, all expressed relative to the agent's current frame. The **action space** is discrete, covering movement in eight directions, sliding, passing, shooting, sprinting, and standing still. The **reward function** follows a sparse setting, where agents receive +100 for winning the match and -1 for losing, without intermediate shaping rewards.

For brevity, several GRF scenarios are referred to using abbreviated names. Specifically, academy_3_vs_2 refers to academy_3_vs_1_with_keeper, academy_2_vs_2 to academy_run_pass_and_shoot_with_keeper, academy_counterattack to academy_counterattack_hard, and academy_4_vs_3 to academy_4_vs_2_with_keeper in the original GRF environment. As shown in Fig. 10, we design full-field variants of these scenarios by repositioning ally and enemy players to opposite half of the court, increasing the difficulty by requiring long-horizon planning and coordinated movement across larger distances. Table 4 provides an overview of the unit configurations and corresponding environment dimensions.

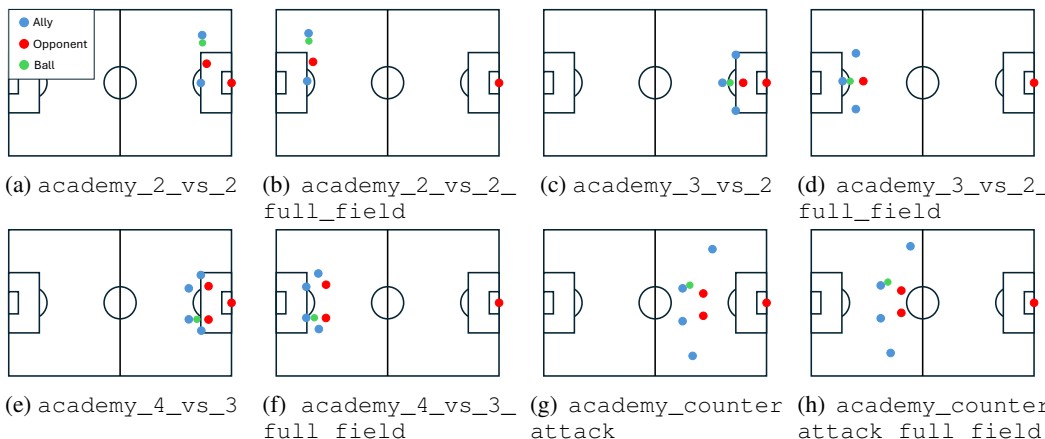

(a) academy_2_vs_2

(b) academy_2_vs_2_full_field

(c) academy_3_vs_2

(d) academy_3_vs_2_full_field

(e) academy_4_vs_3

(f) academy_4_vs_3_full_field

(g) academy_counter attack

(h) academy_counter attack_full_field

Figure 10: Visualization of initial agent positions in GRF scenarios.

| Scenario | Ally | Opponent | State Dim | Obs Dim | Action Dim |
|---|---|---|---|---|---|
| academy_2_vs_2 | 2 center back | 1 goalkeeper, 1 center back | 22 | 22 | 19 |
| academy_2_vs_2 _full_field | 2 center back | 1 goalkeeper, 1 center back | 22 | 22 | 19 |
| academy_3_vs_2 | 3 central midfield | 1 goalkeeper, 1 center back | 26 | 26 | 19 |
| academy_3_vs_2 _full_field | 3 central midfield | 1 goalkeeper, 1 center back | 26 | 26 | 19 |
| academy_4_vs_3 | 4 central midfield | 1 goalkeeper, 2 center back | 34 | 34 | 19 |
| academy_4_vs_3 _full_field | 4 central midfield | 1 goalkeeper, 2 center back | 34 | 34 | 19 |
| academy _counterattack | 1 central midfield, 1 left midfield, 1 right midfield, 1 central front | 1 goalkeeper, 2 center back | 34 | 34 | 19 |
| academy _counterattack _full_field | 1 central midfield, 1 left midfield, 1 right midfield, 1 central front | 1 goalkeeper, 2 center back | 34 | 34 | 19 |

Table 4: GRF scenario configuration

# F EXPERIMENTAL DETAILS

FIM is implemented on top of the open-source framework from (Hu et al., 2021), which is also used to run QMIX (Rashid et al., 2018) and QPLEX (Wang et al.). LAIES (Liu et al., 2023), RODE (Wang et al., 2021b), MASER (Jeon et al., 2022), CDS (Li et al., 2021a), and FoX (Jo et al., 2024) are evaluated using the original code and settings provided by their respective authors. Experiments are conducted on an NVIDIA RTX 3090 GPU with an Intel Xeon Gold 6348 CPU (Ubuntu 20.04). Training completes within two days for Push-2-Box and SMAC, while each GRF scenario requires less than two days to reach 5 million timesteps. We begin by describing the baseline algorithms in Appendix F.1, outline the hyperparameter setup of FIM in Appendix F.2, and conclude with visualizations of entropy and CoG state dimension selection in Appendix F.3.

## F.1 DETAILED DESCRIPTION OF BASELINE ALGORITHMS

- **QMIX** (Rashid et al., 2018) factorizes the joint action-value into individual utilities combined by a monotonic mixing network, ensuring consistency between global and individual greedy actions. Code: https://github.com/hijkzzz/pymarl2

- **QPLEX** (Wang et al.) extends QMIX with a duplex dueling architecture, decomposing joint value into individual value and advantage while enforcing the IGM principle. Code: https://github.com/hijkzzz/pymarl2

- **LAIES** (Liu et al., 2023) incentivizes agents to influence external task-relevant states via intrinsic rewards for both individual and joint impacts. Code: https://github.com/liuboyin/LAIES

- **RODE** (Wang et al., 2021b) employs hierarchical role-based policies where agents periodically select roles to guide low-level actions, enabling scalable specialization. Code: https://github.com/TonghanWang/RODE

- **MASER** (Jeon et al., 2022) enhances exploration by assigning subgoals from past trajectories, rewarding agents for revisiting informative states. Code: https://github.com/Jiwonjeon9603/MASER

- **CDS** (Li et al., 2021a) encourages policy diversity under parameter sharing by maximizing mutual information between agent identity and trajectory. Code: `https://github.com/lich14/CDS`

- **FoX** (Jo et al., 2024) promotes structured exploration by maximizing entropy of agent formations and their mutual information with team structure. Code: `https://github.com/hyeon1996/FoX`

- **COMA** (Foerster et al., 2018b) is a counterfactual multi-agent policy gradient method that assigns credit using a centralized critic with counterfactual baselines. Code: `https://github.com/oxwhirl/pymarl2`

## F.2 HYPERPARAMETER SETUP OF THE PROPOSED FIM

| Scenario | $\eta$ | $\alpha$ | $\delta$ | $\kappa$ |
|---|---|---|---|---|
| Push-2-Box | 5 | 0.1 | 0.1 | 0 |
| **Starcraft Multi-agent Challenge (Sparse)** | | | | |
| `5m_vs_6m` | 50 | 5 | 0.05 | 0.01 |
| `8m_vs_9m` | 50 | 5 | 0.05 | 0.01 |
| `3s_vs_5z` | 50 | 1 | 0.1 | 0.005 |
| `corridor` | 50 | 5 | 0.05 | 0.01 |
| `MMM2` | 50 | 5 | 0.15 | 0.01 |
| `6h_vs_8z` | 50 | 5 | 0.1 | 0.05 |
| `27m_vs_30m` | 50 | 5 | 0.05 | 0.01 |
| `3s5z_vs_3s6z` | 50 | 5 | 0.15 | 0.01 |
| **Google Research Football (Sparse)** | | | | |
| `academy_2_vs_2` | 10 | 1 | 0.5 | 0.01 |
| `academy_2_vs_2_full_field` | 10 | 10 | 0.5 | 0.01 |
| `academy_3_vs_2` | 10 | 10 | 0.1 | 0.01 |
| `academy_3_vs_2_full_field` | 10 | 1 | 0.1 | 0.01 |
| `academy_4_vs_3` | 10 | 10 | 0.2 | 0.01 |
| `academy_4_vs_3_full_field` | 10 | 10 | 0.2 | 0.01 |
| `academy_counterattack` | 10 | 10 | 0.1 | 0.01 |
| `academy_counterattack_full_field` | 10 | 1 | 0.5 | 0.001 |
| **Starcraft Multi-agent Challenge v2 (Sparse)** | | | | |
| `protoss_5_vs_5` | 50 | 5 | 0.25 | 0.01 |
| `terran_5_vs_5` | 50 | 5 | 0.25 | 0.01 |
| `zerg_5_vs_5` | 50 | 5 | 0.25 | 0.01 |
| **Petting Zoo Multi Particle Environments** | | | | |
| `simple_spread_v3` | 50 | 5 | 0.25 | 0.01 |

Table 5: Scenario specific hyperparmeter setup of FIM

| Hyperparameters | Value |
|---|---|
| Optimizer | Adam |
| $\epsilon$ anneal step | 50000 |
| Replay buffer size | 5000 |
| Target update interval | 200 |
| Mini-batch size | 32 |
| Mixing network dim | 32 |
| Discount factor $\gamma$ | 0.99 |
| Learning rate | 0.0005 |
| Dynamics model $\hat{s}(\cdot)$ layer | 3 |
| Dynamics model $\hat{s}(\cdot)$ dim | 128 |

Table 6: Common hyperparameter setting of FIM

The default hyperparameter settings of FIM, which are generally shared across scenarios, are summarized in Table 6. Scenario-specific tuning of the trace scaling factor $\eta$, intrinsic reward weight $\alpha$, entropy threshold $\delta$, and influence threshold $\kappa$ is provided in Table 5, while the trace ceiling $c_{\max}$ is fixed at 10, the softmax temperature in $\mathrm{Softmax}(-\mathcal{H}(d))$ is set to 0.1, and the trace decay factor $\lambda$ is fixed at 0.95 across all scenarios.

## F.3 VISUALIZATION OF ENTROPY $\mathcal{H}(d)$ AND COG STATE DIMENSION SELECTION

Fig. 11 and Fig. 12 visualize $\mathcal{H}(d)$ for SMAC and GRF. To facilitate comparison, $\mathcal{H}(d)$ values are min-max normalized to the range $[0,1]$ within each environment. In GRF, $\mathrm{CoG}_\delta$ consistently highlights goalkeeper positions, which are critical for evaluating offensive positioning and shot opportunities, as discussed in Appendix H. In SMAC, it emphasizes enemy health, a key factor for prioritizing targets and coordinating attacks. Although ally-specific features such as unit type, which only change when an agent is eliminated by enemy, are included in the CoG dimensions, FIM emphasizes features that allies can directly influence and thus prioritizes enemy health and shield to increase influence eligibility traces.

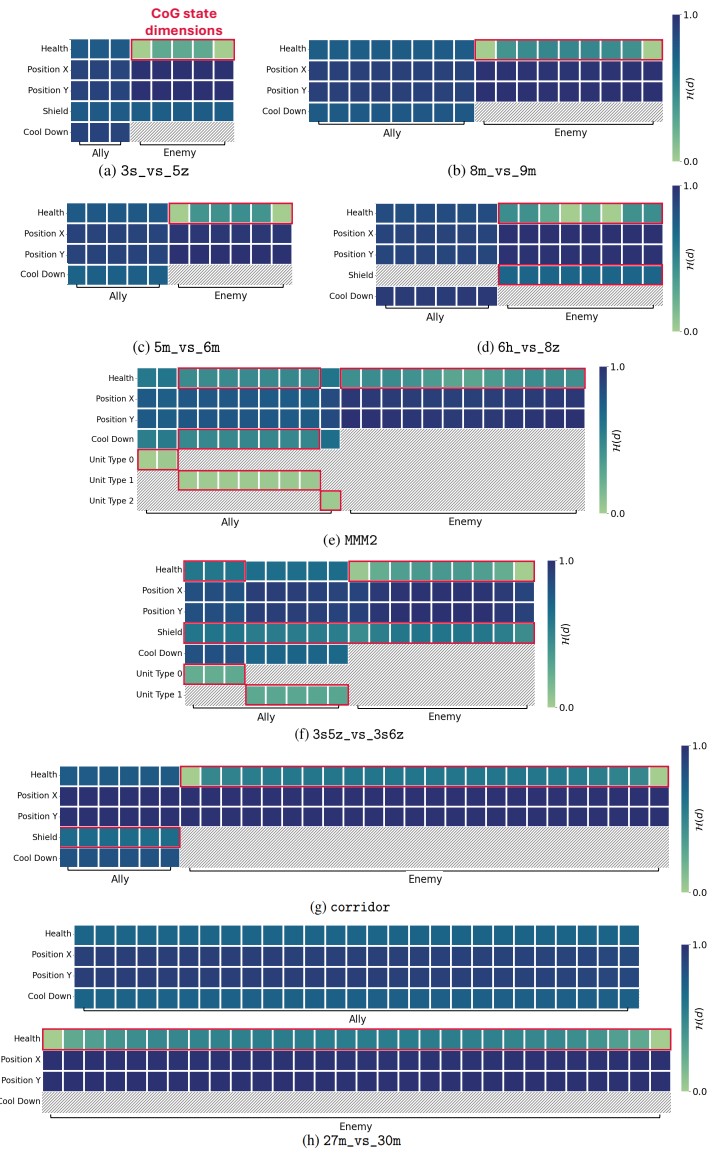

Figure 11: SMAC $\mathcal{H}(d)$ visualization

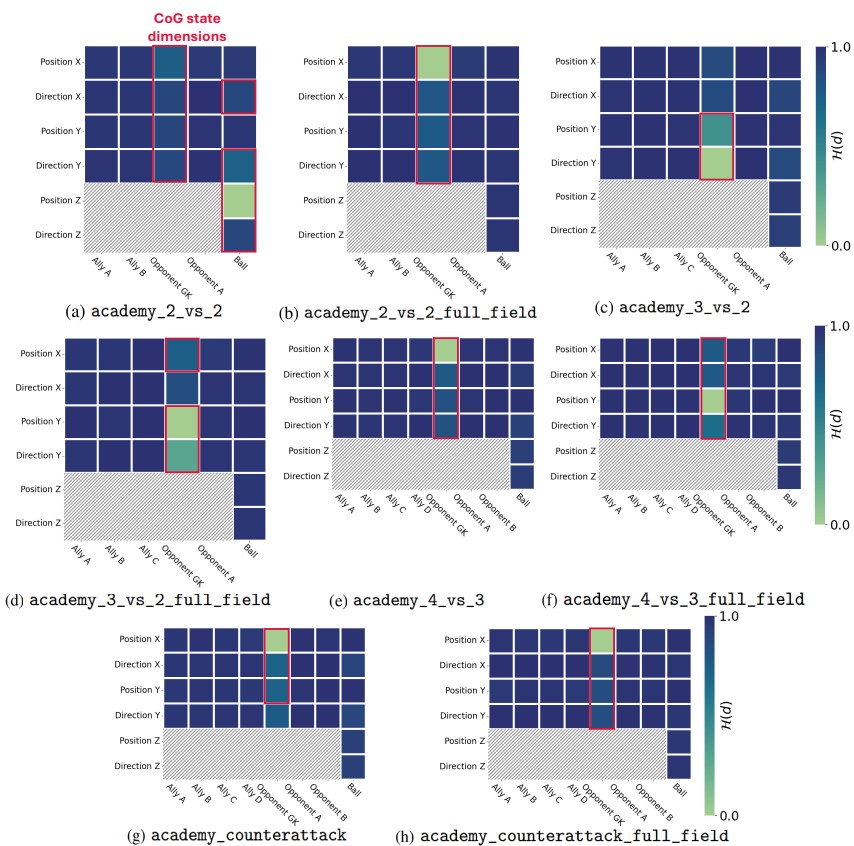

Figure 12: GRF $\mathcal{H}(d)$ visualization

# G   COMPARISON OF COMPUTATIONAL COMPLEXITY

FIM computes intrinsic rewards by estimating each agent's influence through counterfactual marginalization over its action set $\mathcal{A}$ for every dimension in $\text{CoG}_\delta$. This results in a space complexity of $O(|\mathcal{N}| \cdot |\mathcal{A}| \cdot |\text{CoG}_\delta|)$ per timestep, while the time complexity remains $O(1)$ due to GPU parallelization. FIM uses a lightweight three-layer multilayer perceptron (MLP) as the forward transition model and does not alter the main Q-network architecture, keeping computational overhead minimal. We compare FIM against QMIX with dense rewards (QMIX-DR), since sparse-reward QMIX often converges to tie-seeking behaviors that avoid conflict (Liu et al., 2023), resulting in minimal policy updates and unrealistically low computational cost. As shown in Table 7, FIM's average computation time per 1 million timesteps is comparable to QMIX-DR. In 3s5z_vs_3s6z, FIM also requires fewer timesteps to reach a 60% success rate, demonstrating strong efficiency. Even in high-dimensional scenarios such as 27m_vs_30m, FIM maintains computational costs comparable to QMIX-DR, indicating that the added influence modeling does not introduce significant overhead. These results emphasize FIM's ability to enhance agent behavior without compromising computational cost.

| Scenario | FIM | QMIX-DR |
|---|---|---|
| 3s_vs_5z | 72.40 min
5.73M | 70.65 min
4.21M |
| 3s5z_vs_3s6z | 126.43 min
8.26M | 123.23 min
13.05M |
| 27m_vs_30m | 155.78 min
14.36M | 139.06 min
3.47M |

Table 7: Average computation time (in minutes) per 1 million timesteps (top row) and the number of timesteps (in millions) required to reach a 60% success rate (bottom row).

## H    TRAJECTORY ANALYSIS IN GRF

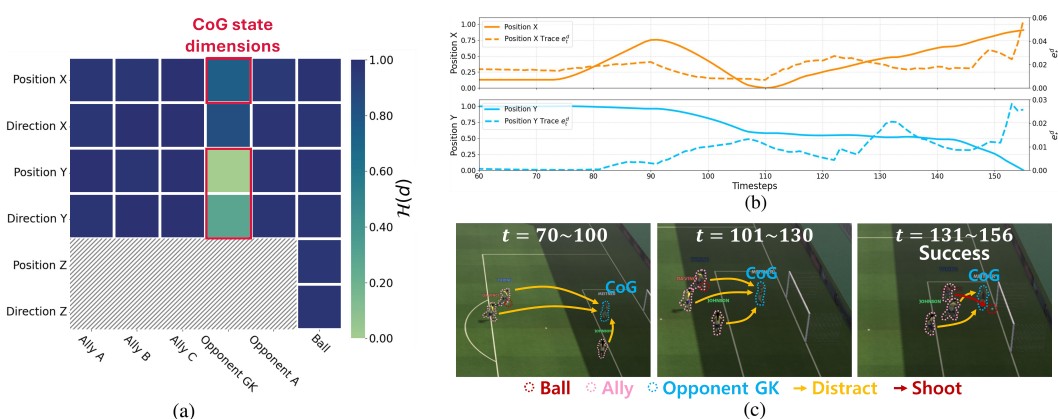

(a)                                                      (b)

(c)

Figure 13: FIM trajectory in GRF `academy_3_vs_2_full_field`

In GRF, the CoG state dimensions identified by FIM primarily correspond to the position of the opponent goalkeeper, as illustrated in Fig. 13(a). These dimensions exhibit low entropy under initial behavior policy, since the goalkeeper typically remains stationary and only shifts position when a ball-carrying agent approaches the goalpost. This characteristic makes the goalkeeper's state both stable and strategically significant, as displacing it creates scoring opportunities and thus serves as a valuable proxy objective in sparse reward settings. Accordingly, FIM guides agents to influence the goalkeeper's position.

As shown in Fig. 13(b)-(c), this insight is reflected in the agent trajectory. Around $t \approx 70$, agents begin to receive intrinsic rewards by subtly influencing the goalkeeper's position, even while positioned far from the goal area. By $t \approx 100$, the accumulated eligibility traces further incentivize agents to continue exerting influence over the goalkeeper, enabling a gradual progression toward the goal. Near $t \approx 130$, the goalkeeper briefly moves out of position, and the attacking agent capitalizes on this opportunity to score. Notably, FIM guides agents to approach the goal proactively and maintain persistent influence over the goalkeeper's positioning, which serves as a task critical factor for successful coordination in this environment, particularly under sparse reward conditions.

## I    ADDITIONAL EXPERIMENTS

In this section, we present additional experiments that further validate the generality, robustness, and interpretability of our proposed framework. First, we evaluate FIM across different cooperative MARL benchmarks, including SMACv2 and MPE, in Appendix. I.1. Next, we provide extended ablation studies that analyze the independent contributions of SFI and AFI under various settings in Appendix. I.2. We then examine the dynamic update of CoG dimensions and show that the entropy-based selection remains adaptive as the behavior policy evolves during training in Appendix. I.3. Finally, we investigate how the components of FIM interact with the LAIES in Appendix. I.4.

### I.1    GENERALITY OF FIM ACROSS SMACv2 AND MPE

To assess the generality of FIM across diverse cooperative MARL settings, we extended our experiments to SMACv2 (Ellis et al., 2023), which introduces richer unit types and randomized initial configurations compared to the original SMAC. We conducted experiments on three representative scenarios (`terran_5_vs_5`, `zerg_5_vs_5`, and `protoss_5_vs_5`) under the fully sparse reward setting. As shown in Fig. 14, the CoG dimensions emphasize enemy-related features such as health and shield, which are critical for focusing fire and coordinating attacks. Although ally-specific features (e.g., unit type) also exhibit low variability, FIM prioritizes enemy features from which collaborative influence yields greater intrinsic rewards. Consequently, as reported in Fig. 15, FIM achieves strong performance across all scenarios, outperforming recent baselines and even surpassing QMIX with dense rewards (QMIX-DR).

We further evaluated FIM in PettingZoo MPE benchmark (Terry et al., 2021) `simple_spread_v3` which is highly dynamic environment. In this task, positions, veloci-

ties, and relative features evolve continuously, leaving no trivially stable dimensions. Nevertheless, as shown in Fig. 16(a), FIM was able to identify relatively stable dimensions by leveraging entropy differences. Since actions directly control velocity, agent velocities fluctuate heavily even under individual actions, leading to high entropy. In contrast, positions and landmark-relative positions change more gradually unless velocity is consistently applied in the same direction, resulting in lower entropy. Position dimensions, crucial for target-approaching behavior, are therefore selected as CoG. Fig. 16(b) presents test return comparison, showing FIM converges faster to higher return value compared to QMIX.

A key reason for this generality is that CoG dimensions represent state variables less affected by uncoordinated actions, and thus mark regions that are hard to influence without cooperation. While not always direct task termination indicators, they highlight underexplored aspects of the environment that require joint effort. FIM rewards agents for influencing these dimensions, steering exploration toward coordination-critical regions. For instance, in GRF the goalkeeper state is often selected as CoG: although not itself the goal signal, coordinating to disrupt it improves scoring. This illustrates how CoG dimensions, even if not directly tied to objectives, can guide agents toward meaningful cooperation, explaining the generality of FIM across SMACv2, MPE, and beyond.

| State Dim Group | Average $\mathcal{H}(d)$ |
|---|---|
| Ally Health | $0.26 \pm 0.02$ |
| Ally CoolDown | $0.16 \pm 0.03$ |
| Ally Position | $0.40 \pm 0.01$ |
| Ally Unit Type | $0.01 \pm 0.01$ |
| Enemy Health | $0.19 \pm 0.00$ |
| Enemy Position | $0.97 \pm 0.02$ |
| Enemy Unit Type | $0.01 \pm 0.01$ |

(a) `zerg_5_vs_5`

| State Dim Group | Average $\mathcal{H}(d)$ |
|---|---|
| Ally Health | $0.22 \pm 0.01$ |
| Ally CoolDown | $0.30 \pm 0.01$ |
| Ally Position | $0.49 \pm 0.20$ |
| Ally Unit Type | $0.01 \pm 0.01$ |
| Enemy Health | $0.11 \pm 0.00$ |
| Enemy Position | $0.97 \pm 0.03$ |
| Enemy Unit Type | $0.01 \pm 0.01$ |

(b) `terran_5_vs_5`

| State Dim Group | Average $\mathcal{H}(d)$ |
|---|---|
| Ally Health | $0.19 \pm 0.01$ |
| Ally CoolDown | $0.34 \pm 0.10$ |
| Ally Position | $0.44 \pm 0.04$ |
| Ally Shield | $0.15 \pm 0.02$ |
| Ally Unit Type | $0.01 \pm 0.01$ |
| Enemy Health | $0.04 \pm 0.00$ |
| Enemy Position | $0.98 \pm 0.01$ |
| Enemy Shield | $0.18 \pm 0.00$ |
| Enemy Unit Type | $0.01 \pm 0.01$ |

(c) `protoss_5_vs_5`

Figure 14: SMACv2 $\mathcal{H}(d)$ visualization

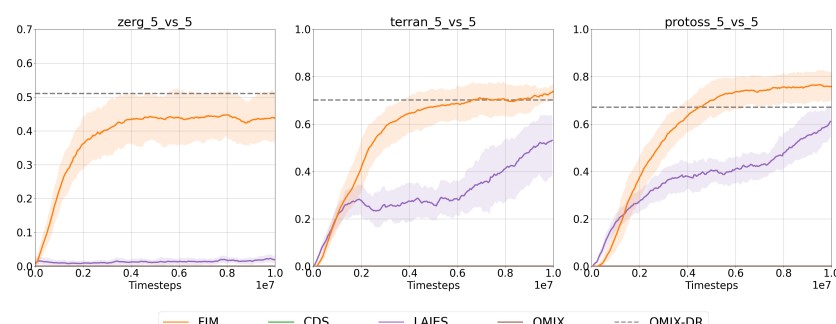

Figure 15: Performance comparison on SMACv2 environments

| State Dim Group | Average $\mathcal{H}(d)$ |
|---|---|
| Agent Velocity | $0.49 \pm 0.10$ |
| Agent Position | $0.18 \pm 0.14$ |
| Landmark Rel. Pos. | $0.18 \pm 0.01$ |
| Inter-Agent Dist. | $0.79 \pm 0.16$ |

(a) $\mathcal{H}(d)$ visualization

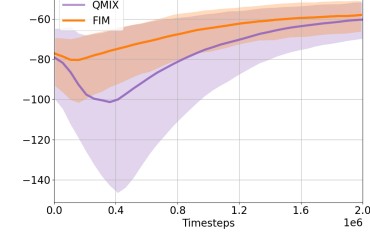

(b) Performance comparison

Figure 16: Experiment on PettingZoo MPE `simple_spread_v3`

## I.2 EXTENDED ANALYSIS ON ABLATION STUDIES

To further evaluate the robustness of FIM, we conduct experiments on four challenging scenarios: SMAC `3s_vs_5z`, SMAC `3s5z_vs_3s6z`, GRF `academy_3_vs_2`, and GRF `academy_3_vs_2_full_field`. Our analysis focuses on four aspects: (i) alternatives to the SFI state selection mechanism, (ii) the impact of each module in FIM through a component ablation study, (iii) the effect of varying the trace scaling factor $\eta$, (iv) the sensitivity to the reward scaling factor $\alpha$, and (v) the role of the trace decay factor $\lambda$ in the trace mechanism.

**Alternatives to the SFI State Selection Mechanism**

We investigate alternative strategies to the SFI state selection mechanism for determining the set of state dimensions to influence: *no-state-selection*, *external state focusing influence* (EFI), and *least-change state focusing influence* (LFI). In all cases, the intrinsic reward is computed as in FIM, with each variant differing only in the selection of the state dimension set $\mathcal{D}$. The chosen dimensions for each variant are summarized in Table 8. The no-state-selection variant sets $\mathcal{D}$ to include all state dimensions, effectively applying no filtering. EFI manually selects task-relevant external features, following the approach of LAIES (Liu et al., 2023): enemy health, shield, and positions in SMAC; and opponent and ball positions and directions in GRF. LFI selects the $n = |\mathrm{CoG}_\delta|$ state dimensions with the smallest average temporal change $|s_{t+1}^d - s_t^d|$ under a initial behavior policy. In SMAC, this typically includes enemy health and ally positions, while in GRF, it often selects ally direction features due to their relatively small-scale temporal changes.

As shown in Fig. 17, while some SFI variants show comparable performance in (a) and (c), the state dimensions selected by SFI consistently lead to the highest overall performance. When variants include easily influenced features such as ally position, agents tend to exploit these trivial dimensions, leading to reward hacking and suboptimal behavior. These findings underscore the effectiveness of FIM's entropy-based selection, which identifies stable and causally meaningful CoG dimensions to promote more coordinated and purposeful agent behavior.

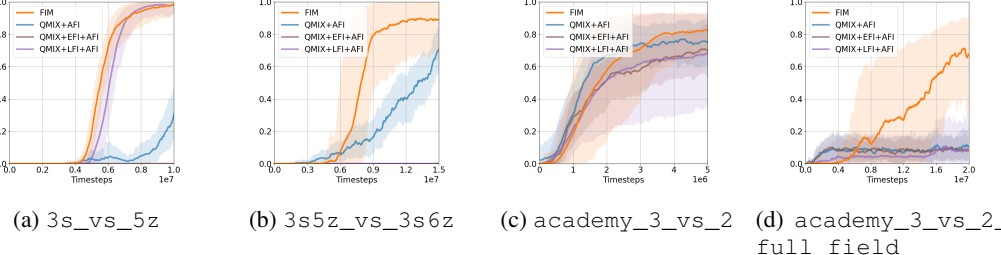

(a) `3s_vs_5z`  (b) `3s5z_vs_3s6z`  (c) `academy_3_vs_2`  (d) `academy_3_vs_2_full_field`

Figure 17: Alternatives to SFI

| Scenario | SFI | EFI | LFI |
|---|---|---|---|
| `3s_vs_5z` | enemy health | enemy health, enemy shield, enemy position | enemy health |
| `3s5z_vs_3s6z` | enemy health, enemy shield, ally health, ally shield, ally unit type | enemy health, enemy shield, enemy position | enemy health, enemy position, ally position |
| `academy_3_vs_2` | goalkeeper position, goalkeeper direction | opponent position, opponent direction, ball position, ball direction | ally direction |
| `academy_3_vs_2_full_field` | goalkeeper position, goalkeeper direction | opponent position, opponent direction, ball position, ball direction | goalkeeper direction, ally direction |

Table 8: Selected state dimensions comparison for SFI, EFI and LFI

**Component Evaluation**

Fig. 18 compares four variants: vanilla QMIX, QMIX with state focusing influence (SFI), QMIX with agent focusing influence (AFI), and the full FIM framework that integrates both components. In SMAC, focused fire emerges as a key cooperative strategy, where agents coordinate to attack a single enemy unit at a time. SFI supports this behavior by directing influence toward task-relevant CoG dimensions, such as enemy health and shield, while AFI encourages agents to maintain consistent attention across time. Although each component improves performance on its own, only their combination in FIM reliably induces and sustains focused fire, resulting in the highest success rates, as shown in Fig. 18(a)-(b). A similar effect is observed in GRF, where SFI identifies the goalkeeper's position as a key dimension, and AFI ensures that agents continue to influence it over multiple steps in order to exploit brief chance when the goalkeeper is out of position. Together, these components enable coordinated behaviors that consistently outperform all other variants, as shown in Fig. 18(c)-(d).

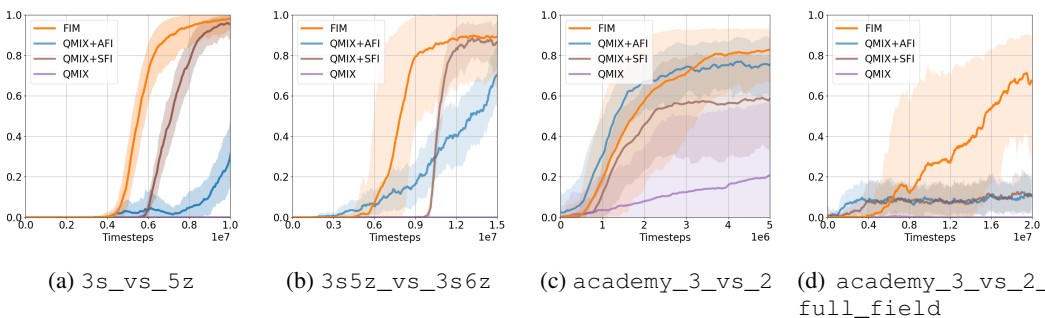

(a) 3s_vs_5z  (b) 3s5z_vs_3s6z  (c) academy_3_vs_2  (d) academy_3_vs_2_
full_field

Figure 18: Component evaluation

**$\eta$ Effect Analysis**

We investigate how different settings of the trace scaling factor $\eta$ affect performance by evaluating $\eta \in \{1, 5, 10, 50, 100\}$, as shown in Fig. 19. The results show that the choice of $\eta$ significantly influences learning outcomes such that extreme values on either end tend to impair performance. When $\eta$ is too low, a larger amount of influence over a longer period is required to sufficiently increase the eligibility trace, which may cause the system to become insensitive to recent influence and fail to reflect meaningful credit accumulation. On the other hand, if $\eta$ is too high, the eligibility trace rapidly reaches the ceiling $c_{\max}$, leading to two undesirable effects. First, it reduces the discriminative power between dimensions, as many attain the same maximum eligibility value. Second, it diminishes the incentive for agents to sustain influence across multiple timesteps, since eligibility values remain near the maximum regardless of temporal decay. Based on these findings, we set $\eta = 50$ for SMAC and $\eta = 10$ for GRF, which yielded the most stable and effective performance across tasks.

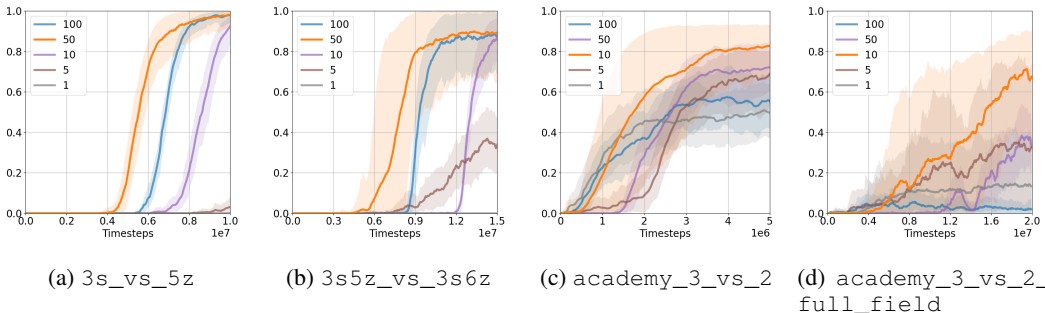

(a) 3s_vs_5z  (b) 3s5z_vs_3s6z  (c) academy_3_vs_2  (d) academy_3_vs_2_
full_field

Figure 19: Effect of $\eta$

$\alpha$ **Effect Analysis**

We examine how the reward scaling factor $\alpha$ affects performance by testing values in $\alpha \in \{0.1, 0.5, 1, 5, 10, 50\}$, as shown in Fig. 20. When $\alpha$ is too small, the intrinsic reward signal becomes negligible, preventing agents from effectively learning the influence-guided strategy promoted by FIM. Conversely, setting $\alpha$ too large causes agents to over-prioritize intrinsic rewards, ignoring critical environmental feedback and converging to suboptimal behaviors. To ensure balanced learning, we select $\alpha$ values that are well aligned with the extrinsic reward scale of each scenario. This balance is particularly important in sparse-reward environments, where intrinsic signals must guide exploration without overwhelming the task objective. Our selected $\alpha$ values thus ensure that agents benefit from influence-driven incentives while still grounding their behavior in task success.

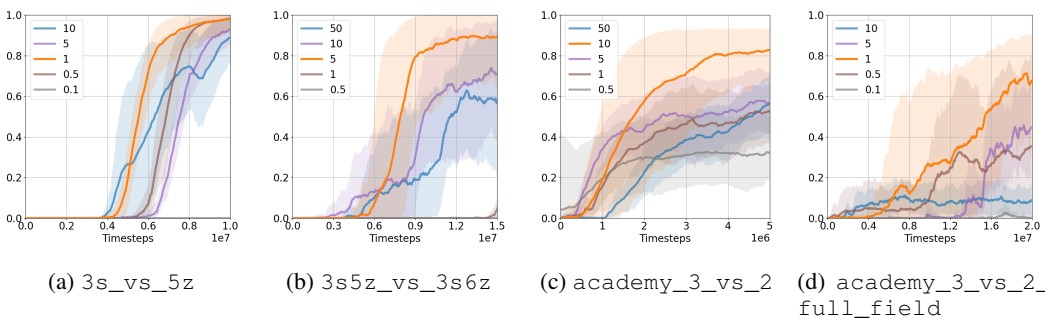

(a) `3s_vs_5z`    (b) `3s5z_vs_3s6z`    (c) `academy_3_vs_2`    (d) `academy_3_vs_2_ full_field`

Figure 20: Effect of $\alpha$

$\lambda$ **Effect Analysis**

We examine the effect of the trace decay factor $\lambda$ by varying it across $\lambda \in \{0.8, 0.85, 0.9, 0.95, 1\}$. The parameter $\lambda$ determines how long the influence of past actions persists in the eligibility trace. As shown in Fig. 21, when $\lambda = 1$, the trace never decays, causing all past influence, whether recent or outdated, to be treated equally. This undermines the ability to prioritize recent, coordinated influence, weakening short-term focus and resulting in suboptimal performance. Conversely, when $\lambda$ is too small, eligibility decays too rapidly, limiting the benefit of temporal accumulation and again degrading learning. Through empirical evaluation, we find that $\lambda = 0.95$ consistently yields the best performance and adopt it as the default across all scenarios.

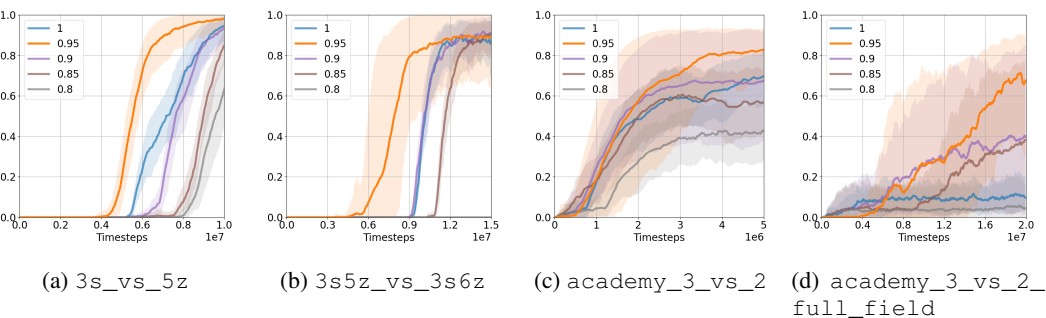

(a) `3s_vs_5z`    (b) `3s5z_vs_3s6z`    (c) `academy_3_vs_2`    (d) `academy_3_vs_2_ full_field`

Figure 21: Effect of $\lambda$

### I.3 DYNAMIC COG UPDATE

To verify that entropy-based CoG selection can adapt to the current behavior policy $\beta$ that keeps evolving, we conduct an additional experiment using the current behavior policy $\beta$ to update CoG dimensions. We recomputes CoG dimensions every 250K timesteps. At each update point, we estimate per-dimension entropy $\mathcal{H}(d)$, derive a new entropy-based weight vector $w_d^{\text{new}} = \text{Softmax}(-\mathcal{H}(d))$, and then smoothly update the CoG weights via $w_d \leftarrow (1-\phi)w_d + \phi w_d^{\text{new}}$ with $\phi$ fixed at $0.05$, ensuring that the intrinsic-reward structure evolves gradually without destabilizing training.

We designed SMAC `3s_vs_5z_shield_100` scenario as a variant of `3s_vs_5z` to evaluate the performance of dynamic CoG updating FIM (abbreviated as dFIM). We increase all enemy shields from 50 to 100 so that an untrained initial policy can never alters enemy health. Consequently, the entropy of enemy-health dimensions is exactly zero at the beginning of training, and these dimensions are therefore excluded from the initial CoG set. This allows us to explicitly test whether the CoG mechanism can recover previously excluded dimensions once agents become capable of affecting them. Fig. 22 shows that enemy-health dimensions, initially absent due to zero entropy, begin to exhibit non-zero temporal variation once agents reliably deplete shield, and are progressively contained as a CoG dimensions. As shown in Fig. 24(a), while FIM that fix CoG dimensions fail to improve performance, dFIM successfully improve performance.

We also tested dFIM in `3s_vs_5z` in which initial policy is able to influence all state dimensions. As shown in Fig. 23, dynamically updating the CoG led to only minor changes, such as the inclusion of a few additional dimensions such as ally features. As shown in Fig. 24(b), dFIM brought little additional benefit compared to using a fixed set. Since the initial policy in most of our main experimental scenarios can similarly influence all dimensions from the outset, we adopt the fixed-CoG version of FIM in the main experiments to keep the overall implementation simple.

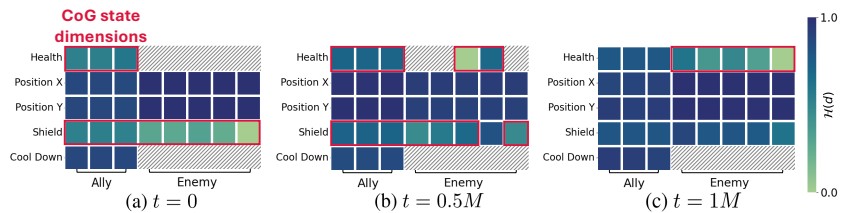

Figure 22: $\mathcal{H}(d)$ at each timestep in `3s_vs_5z_shield_100`

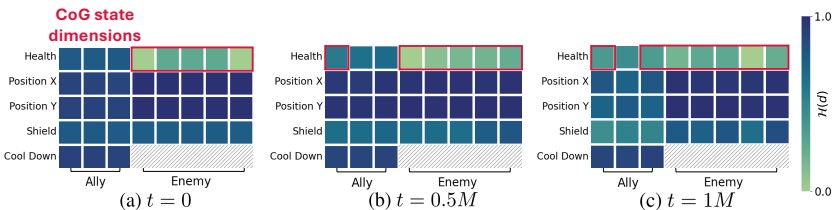

Figure 23: $\mathcal{H}(d)$ at each timestep in `3s_vs_5z`

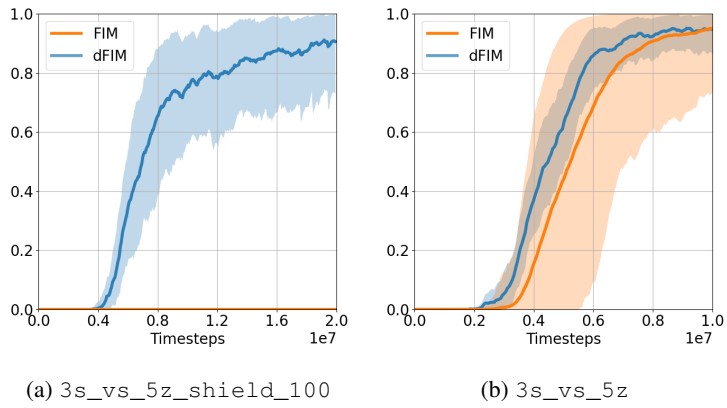

(a) `3s_vs_5z_shield_100`       (b) `3s_vs_5z`

Figure 24: Performance comparison of FIM and dFIM

## I.4 LAIES WITH FIM COMPONENTS

To better understand the individual contributions of selective state focusing (SFI) and accumulated future influence (AFI), we further evaluate how the LAIES framework behaves when augmented with these components. We conduct experiments in SMAC `3s_vs_5z` and `27m_vs_30m`, where vanilla LAIES is known to struggle to make progress. In these experiments, we follow the original LAIES setup and use the full external enemy feature vector exactly as defined in their paper. vanila LAIES attempts to influence the entire enemy feature vector, which becomes problematic in scenarios that require highly focused strategies such as `3s_vs_5z`, or in maps with many enemies such as `27m_vs_30m`, where the number of relevant dimensions is large and the influence signal becomes overly diffuse.

We construct two variants of LAIES: LAIES+SFI, which replaces LAIES's extrinsic state with our CoG-selected dimensions, and LAIES+AFI, which rescales LAIES's influence using per-dimension eligibility traces accumulated over time. As shown in Fig. 25, both modifications improve the performance of LAIES. From the SFI perspective, this demonstrates that concentrating influence on a small number of key dimensions (e.g., enemy health) is more effective than distributing it across the full enemy feature vector. From the AFI perspective, the gains indicate that temporally accumulated influence provides a complementary signal absent in the original LAIES formulation. Together, these results confirm that SFI and AFI play essential and complementary roles, and that each component independently enhances the effectiveness of influence-based exploration.

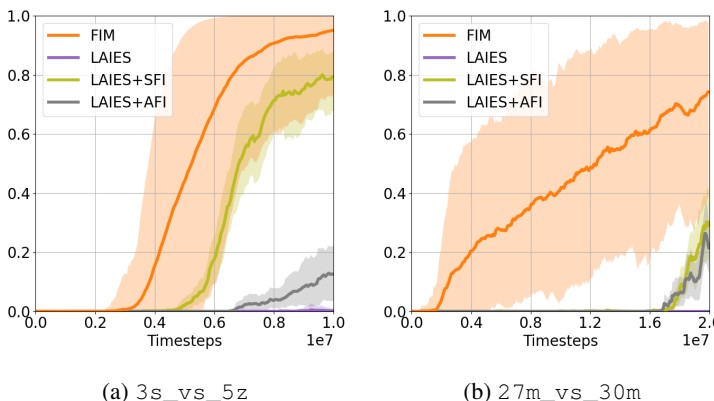

(a) `3s_vs_5z`  (b) `27m_vs_30m`

Figure 25: Performance evaluation of LAIES with FIM components

## J ANALYSIS OF THE LEARNED DYNAMICS MODEL

Since the intrinsic reward in FIM is computed from the predictions of the learned dynamics model $\hat{s}$, its accuracy directly influences the reward signal. While a high mean-squared error (MSE) might seem detrimental, our results suggest that prediction inaccuracies can also serve a constructive role by implicitly encouraging exploration of regions with complex or less predictable dynamics. In this sense, model error may act as a form of curiosity, resonating with ideas from curiosity-driven exploration in model-based RL (Pathak et al., 2017).

To examine this effect empirically, we analyzed the SMAC `3s_vs_5z` scenario. As shown in Fig. 26, the forward model's MSE gradually increased during training, likely reflecting exposure to more diverse transitions. Notably, this trend coincided with a steady improvement in win rate, suggesting that moderate prediction error did not destabilize learning but rather correlated with productive exploration, ultimately supporting performance gains.

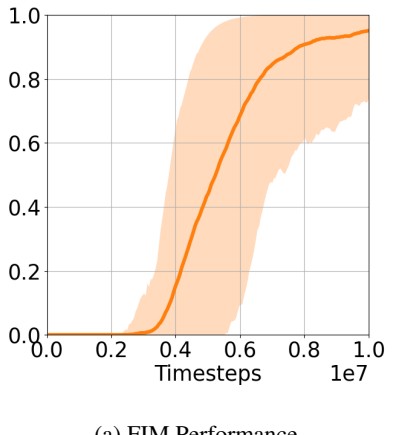

(a) FIM Performance

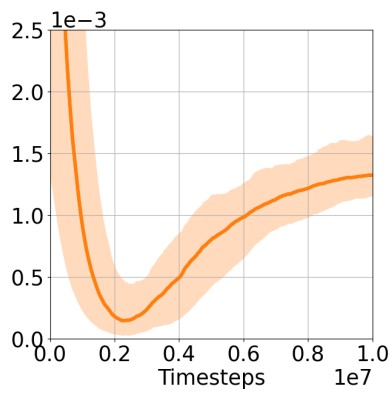

(b) Mean squared error loss of $\hat{s}$

Figure 26: Comparison of FIM performance and mean squared error loss of $\hat{s}$ in `3s_vs_5z`