# OpenReview forum: "Center of Gravity-Guided Focusing Influence Mechanism for Multi-Agent Reinforcement Learning"
_ICLR.cc/2026/Conference — Submitted to ICLR 2026_

### Official Review · Reviewer_Mvru · 2025-10-27

**Soundness:** 3
**Presentation:** 3
**Contribution:** 3
**Rating:** 6
**Confidence:** 4

**Summary:**

This paper introduces the Focusing Influence Mechanism (FIM), a novel framework for cooperative multi-agent reinforcement learning (MARL) in sparse reward environments. Drawing inspiration from Clausewitz's military theory of "Center of Gravity" (CoG), the authors propose a principled approach to identify state dimensions that exhibit low variability under typical agent behaviors but are critical for task completion. The framework consists of two key components: (1) **State Focusing Influence (SFI)**, which uses entropy-based criteria to automatically select CoG dimensions and designs counterfactual intrinsic rewards to guide agents to influence these dimensions, and (2) **Agent Focusing Influence (AFI)**, which employs eligibility traces to maintain persistent and synchronized attention across agents on shared CoG dimensions.

The method is evaluated on three benchmarks: a toy Push-2-Box environment, SMAC (StarCraft Multi-Agent Challenge), and Google Research Football (GRF). Results demonstrate that FIM consistently outperforms strong baselines including QMIX, LAIES, MASER, CDS, FoX, RODE, and QPLEX across all scenarios. The authors provide extensive ablation studies showing the necessity of both SFI and AFI components, as well as analysis of hyperparameter sensitivity.

**Key Contributions:**
1. A novel conceptual framework inspired by military strategy for MARL coordination
2. An entropy-based method for automatically identifying hard-to-change but task-critical state dimensions
3. A eligibility trace mechanism for sustaining coordinated multi-agent attention
4. Comprehensive empirical validation across diverse MARL benchmarks with strong performance gains

**Strengths:**

### 1. **Novel and Well-Motivated Conceptual Framework**
The application of Clausewitz's "Center of Gravity" concept to MARL is creative and well-motivated. The paper clearly articulates why identifying state dimensions that are "stable under typical behaviors but require coordinated effort to change" is important for sparse-reward cooperation. The Push-2-Box example (Figure 1) effectively illustrates the problem and solution.

### 2. **Principled and Automatic State Selection Method**
Unlike prior work (e.g., LAIES) that manually selects task-relevant features, FIM uses an entropy-based criterion to automatically identify CoG dimensions. The normalized temporal change metric (Eq. 3-5) provides a theoretically grounded approach that:
- Accounts for magnitude differences across dimensions through normalization
- Distinguishes between "frequently changing but trivial" vs "rarely but meaningfully changing" dimensions
- Requires no domain knowledge or manual feature engineering

The ablation study (Appendix I) convincingly demonstrates that entropy-based selection outperforms naive alternatives (no-selection, manually-selected EFI, and least-change LFI).

### 3. **Elegant Integration of Eligibility Traces for Multi-Agent Coordination**
The use of eligibility traces to promote persistent and synchronized influence (AFI) is intuitive and effective. The mechanism naturally handles target switching (when a focused object becomes unreachable) and enables sequential commitment to different CoG dimensions. The visualization in Figure 3 and Figure 7(b) clearly shows how this mechanism induces coordinated behaviors like "focus fire" in SMAC.

### 4. **Comprehensive Experimental Validation**
The experimental evaluation is thorough:
- **Diverse benchmarks**: Push-2-Box (toy), SMAC (8 challenging maps), GRF (8 scenarios including full-field variants)
- **Strong baselines**: Compares against 7 recent methods (LAIES, MASER, CDS, FoX, RODE, QPLEX, QMIX-DR)
- **Consistent improvements**: FIM achieves the highest success rates across all environments
- **Detailed analysis**: Includes trajectory visualizations (Fig 7, 13), entropy analysis (Fig 11, 12), and ablation studies
- **Generalization**: Extended evaluation on SMACv2 and MPE (Appendix H) demonstrates broader applicability

**Weaknesses:**

### 1. **Strong Dependence on Initial Policy Quality (Critical Issue)**
The most significant limitation is that CoG dimensions are identified using only 100K episodes from an **initial random policy** and then **remain fixed** throughout training. This design choice has several concerning implications:

**a) Exploration Bias:** If the initial policy fails to explore certain regions or trigger specific state transitions, dimensions that are actually critical may be missed entirely or assigned zero entropy (and thus excluded from CoGδ). For example:
- In a complex environment with conditional mechanics (e.g., "enemy shields only activate when health drops below 50%"), if the initial policy never reduces enemy health sufficiently, shield values would appear static and be excluded.
- This creates a chicken-and-egg problem: you need good exploration to identify important dimensions, but you need to identify important dimensions to guide exploration effectively.

**b) Distribution Shift:** As training progresses and agents learn better policies, the state visitation distribution changes dramatically. Dimensions that were hard-to-change initially may become easy later, and vice versa. The fixed CoG cannot adapt to this shift. While the authors acknowledge this limitation (Section 4.2, line 256-258), they do not address it in their experiments.

**c) Limited Evidence for Dynamic Settings:** The authors claim the framework can be extended to dynamic CoG updates but provide no experimental validation. Given that all tested environments (SMAC, GRF) have relatively static task objectives, it remains unclear whether FIM would work in domains where critical dimensions genuinely evolve over time (e.g., multi-stage tasks, curriculum learning scenarios).

**Suggested Improvement:** The paper would be significantly strengthened by:
- Ablation study showing impact of using policies at different training stages for CoG estimation
- At least one experiment with periodic CoG re-estimation (e.g., every 500K timesteps)
- Analysis of how CoG dimensions change (or remain stable) when estimated at different points

**Questions:**

1. **Initial Policy Bias and Dynamic CoG:**
   - What happens if the initial random policy never discovers certain critical state transitions? How would you detect and recover from such failures?
   - Have you tried estimating CoG at different training stages (e.g., after 500K, 1M, 2M timesteps)? How much do the selected dimensions change?
   - Can you provide experimental validation of the claimed capability to "periodically update β" (line 256-258)? Even a single experiment demonstrating this would strengthen the paper significantly.

2. **Hyperparameter Sensitivity and Transfer:**
   - Can you provide principled guidelines for setting δ, α, and η in new environments? For example, should δ be normalized by the average entropy across all dimensions?
   - How many hyperparameter configurations did you try per environment? How does this compare to the tuning cost of baselines?
   - Have you tested FIM on a completely new environment (not in the paper) to assess transfer difficulty?

3. **Marginal vs Conditional Entropy:**
   - Can you provide analysis or empirical evidence showing when marginal distribution p(Δd) is a good approximation of conditional p(Δd | st)?
   - In GRF, goalkeeper position is selected as CoG, but presumably it only matters when agents are near the goal. Does the marginal approximation adequately capture this context-dependence?

4. **Comparison with LAIES:**
   - In your experiments, did you use the original external state features from LAIES's paper, or did you select features yourself? Please clarify in the revision.
   - Why does LAIES fail on 27m_vs_30m and corridor specifically? Can you provide empirical analysis (e.g., which features did LAIES influence, and why was that suboptimal)?
   - What is the result of using FIM's CoG selection with LAIES's influence mechanism (without AFI)?

---

> ### Author Response · Authors · 2025-11-20
> **Response to Reviewer Mvru**
>
> Thank you very much for your detailed and constructive feedback. Your comments greatly improved the clarity of our work.
>
> **Weakness 1/Question 1 (Dynamic CoG dimension selection)**
>
> We thank the reviewer for the helpful comments regarding the fixed CoG set. As noted in Section 4.2, our framework does not require CoG dimensions to remain static. They can be updated as the policy evolves and the task or the policy causes shifts in which dimensions are critical. This matters when the initial random behavior policy does not sufficiently excite important state dimensions.
>
> Following the reviewer’s suggestion, we implemented an online CoG update in Appendix I.3, recomputing CoG every 250K timesteps under the evolving behavior policy. In ordinary setting such as 3s_vs_5z, the changes were minor (e.g., adding a few dimensions) and provided little benefit over the fixed version. Thus, we use a fixed CoG in the main experiments to keep the implementation simple and avoid additional nonstationarity caused by frequently changing intrinsic reward targets. To verify FIM in more complex scenario, we modified 3s_vs_5z such that enemy health is initially uninfluenceable and thus absent from the initial CoG, the dynamic CoG update gradually incorporates these newly controllable dimensions and enables the task to be solved.
>
> **Question 2 (Hyperparameter sensitivity and transfer)**
>
> We thank for the question on hyperparameter selection. As shown in Fig. 8, we conducted a broad grid search over state selection threshold $\delta$, the trace scaling factor $\eta$, and the intrinsic reward weight $\alpha$, from which we distilled simple guidelines that generalize across tasks. We compute normalized entropies $\mathcal{H}(d)$ and choose $\delta$ slightly below their average, so that the CoG dimensions correspond to clearly lower entropy components without being restricted to only a few extreme outliers. For $\alpha$, we search over ${0.1,0.5,1,5,10,50}$ so intrinsic rewards match the extrinsic scale. The parameter $\eta$ controls trace accumulation, and values such as $10$ or $50$ provide stable temporal growth without abrupt influence spikes.
>
> To test transferability, following the reviewer’s suggestion, we applied FIM to PettingZoo MPE simple_spread_v3 [R.5] in Appendix I.1 using the same rules: set $\delta$ near half the average entropy, fix $\eta=50$, and tune only $\alpha$. FIM again outperformed QMIX, indicating that the method is not overly sensitive and that these heuristics are robust. We thank the reviewer for motivating this clarification.
>
> [R.5] J. Terry et al., PettingZoo, 2021.
>
> **Question 3 (Marginal vs. conditional entropy)**
>
> We thank the reviewer for asking when marginal entropy approximates conditional entropy. Since both are averaged over the same visitation distribution of state, the marginal entropy can be viewed as a mixture of conditional entropies weighted by how often each state is visited. In Appendix D.1, we measured both in GRF academy\_2\_vs\_2 and found that they are numerically close and, importantly, yield nearly identical rankings across dimensions. Because CoG depends only on this ranking, marginal entropy is a reliable and statistically cheaper surrogate.
>
> A concrete example arises in GRF: the goalkeeper moves meaningfully only when the ball approaches dangerous regions. Thus, its position generally has very low entropy under the behavior policy. CoG correctly marks this as a difficult-to-influence dimension, encouraging agents to generate coordinated trajectories that genuinely threaten the goal. This leads to strategic behaviors such as distracting or pulling the goalkeeper out of position, which significantly increases scoring chances.
>
> **Question 4 (Disentangling SFI and AFI contributions)**
>
> We appreciate the reviewer’s questions regarding LAIES and our SFI/AFI components. LAIES uses the full enemy feature vector as its target state, and we follow this configuration. However, influencing the entire vector becomes ineffective in tasks requiring focused strategies (e.g., 3s_vs_5z) or with many enemies (e.g., 27m_vs_30m, corridor), where the influence signal becomes diffuse. Our SFI selects a small set of low-entropy, task critical dimensions, while AFI strengthens influence on these dimensions through temporal accumulation.
>
> To separate their effects, Fig. 8(a) shows that adding either SFI or AFI to QMIX solves 3s_vs_5z, while QMIX and LAIES fail. For a more direct comparison, we applied SFI and AFI to LAIES in 27m_vs_30m and 3s_vs_5z (Appendix I.4). Vanilla LAIES barely progresses, whereas LAIES+SFI and LAIES+AFI both show substantial gains. SFI demonstrates the benefit of focusing on key features such as enemy health; AFI provides a complementary temporal signal. Together, these results highlight the essential and synergistic roles of SFI and AFI.
>
>
> Thank you again for your thoughtful and constructive feedback. Your comments significantly improved the clarity and strength of the paper.

---

### Official Review · Reviewer_XY1P · 2025-10-31

**Soundness:** 3
**Presentation:** 3
**Contribution:** 3
**Rating:** 6
**Confidence:** 4

**Summary:**

reinforcement learning in sparse reward settings. FIM selects low-entropy Center of Gravity (CoG) state dimensions for agents to collectively influence via counterfactual intrinsic rewards and eligibility traces. It is evaluated on Push-2-Box, SMAC, and GRF tasks, demonstrating superior performance over standard value decomposition methods (QMIX, QPLEX) and recent methods designed for sparse reward scenarios (LAIES, MASER, CDS, FoX, RODE).

**Strengths:**

1. Principled approach: The paper provides a clear motivation with entropy-based selection that requires no domain knowledge. Push-2-Box serves as an insightful example, and trajectory visualizations reveal interpretable behaviors such as focus-fire and goalkeeper disruption that align with human strategies.
2. Comprehensive experiments: FIM is robustly compared against both general MARL methods and multiple sparse-reward-specific algorithms across diverse benchmarks. The results indicate strong improvements, particularly in extremely sparse settings where existing techniques struggle.
3. Thorough ablations: Experiments carefully demonstrate the necessity of both selective state targeting (SFI) and persistent coordination (AFI), with sensitivity analyses informing the effects of key hyperparameters.

**Weaknesses:**

1. Sparse reward-specific baselines in Push-2-Box: While QMIX is included as the main baseline in Push-2-Box, it is not designed for sparse reward environments. For stronger evidence, the inclusion of additional baselines explicitly tailored for sparse rewards—such as intrinsic motivation or curiosity-driven exploration approaches—would be valuable. This would clarify whether FIM's superiority is general or primarily relative to methods with limited exploration capacity.
2. Formal justification for CoG selection: The assumption that low-entropy dimensions are always task-relevant needs further theoretical grounding. It is intuitive for box position in Push-2-Box, but less so in SMAC scenarios. A formal analysis of when H(d)<δ aligns with critical state features, or oracle-based ablations, would strengthen this claim.
3. Counterfactual baseline comparison: Despite referencing COMA as a key counterfactual approach, direct experimental comparison is lacking. Including results for COMA, and disentangling the effect of state-dimension-level versus action-level rewards, would better quantify FIM's contributions.

**Questions:**

1. Disentangling SFI and AFI contributions: Could you isolate the benefits from selective state targeting and temporal persistence? Ablate the components by combining LAIES features with FIM's eligibility traces, and report performance on representative tasks.
2. The empirical results show that FIM selects low-entropy CoG dimensions (e.g., health and shield in SMAC) as key coordination targets, and significantly boosts team performance. However, could you provide more insights or visualizations into situations where the entropy-based selection might fail to identify truly task-critical dimensions, especially in environments where the most relevant features do not correspond to lowest entropy? Are there cases where FIM focuses on misleading or sub-optimal state dimensions, and how might this affect coordination? Can the CoG selection mechanism be adapted or combined with domain knowledge to improve robustness across highly diverse scenarios?

---

> ### Author Response · Authors · 2025-11-20
> **Response to Reviewer XY1P**
>
> We sincerely thank the reviewer for the thoughtful and constructive feedback. We carefully revised the manuscript and provide detailed responses below.
>
> **Weakness 1 (Additional baselines in Push-2-Box)**
> We thank the reviewer for the suggestion. Following the request, we updated Section 5 and Fig. 4 to include LAIES, CDS, and FoX as representative intrinsic-motivation and exploration baselines. Despite using reward shaping on box position, LAIES still fails to solve Push-2-Box, suggesting that the lack of agent-focusing influence limits its effectiveness. FoX and CDS, which promote diverse trajectories, also fail on this task. These comparisons show that only our method succeeds and highlight the importance of identifying and concentrating influence on key low-entropy dimensions in sparse-reward settings.
>
> **Weakness 2 (Rationale of CoG dimension selection)**
> We appreciate the question regarding why our method focuses on state dimensions that are difficult to influence under the current behavior policy. We have added clarifications to Section 4.2 (CoG) to address this point. In Q-learning based methods, it is important to visit as diverse a set of states as possible in order to improve convergence [R.4]. To encourage such diverse state visitation, we consider an entropy based approach that captures the information content of state transitions. In environments like Push-2-Box, dimensions with low entropy identify regions where the policy induces insufficient exploration, and CoG is designed to prioritize these low-entropy dimensions, promoting more diverse transitions and broader state-space coverage. To theoretically support this intuition, we added Theorem 4.1, which shows that the proposed CoG state-dimension selection is approximately equivalent, for a given state and behavior policy $\beta$, to choosing the dimension whose next-state distribution has the smallest normalized entropy. This result provides a theoretical motivation for our CoG based selection rule, which is designed to encourage more effective state-space exploration.
>
> These entropy based mechanisms are grounded in general information theoretic quantities, so they also operate effectively in standard MARL benchmarks such as SMAC and GRF. We provide corresponding analyses in Section 5.3 and Appendix F.3, where we examine how CoG state dimensions (under explored state dimensions) are selected in both SMAC and GRF.
>
> [R.4] Richard S. Sutton and Andrew G. Barto. Reinforcement Learning: An Introduction, 2nd ed., 2018.
>
> **Weakness 3 (Additional COMA baseline)**
> We thank the reviewer for suggesting additional COMA comparisons. We added COMA results to Fig. 5 and Fig. 6. COMA also struggles in sparse-reward SMAC and GRF tasks, while FIM consistently outperforms it. Since LAIES and FoX already incorporate counterfactual-style intrinsic signals, our original baselines covered methods in this family; the new COMA results further confirm that our improvements stem from the CoG mechanism rather than from missing counterfactual methods.
>
> **Question 1 (Disentangling SFI and AFI contributions)**
> We thank the reviewer for the question. LAIES uses the full enemy feature vector as the target state, and we follow this design. However, influencing all features simultaneously becomes ineffective when tasks require narrow focus (e.g., 3s_vs_5z) or involve many enemies (e.g., 27m_vs_30m). SFI identifies a compact set of task-critical low-entropy dimensions, and AFI amplifies influence on them over time. Fig. 8(a) shows that either component alone solves 3s_vs_5z, while QMIX and LAIES fail. For larger settings (per reviewers Mvru and XY1P), we applied SFI and AFI to LAIES in 27m_vs_30m and 3s_vs_5z (Appendix I.4). Vanilla LAIES barely improves, while LAIES+SFI and LAIES+AFI show clear gains. SFI highlights the value of focusing on key features such as enemy health, and AFI supplies a complementary temporal signal. These results demonstrate that both components play essential and synergistic roles.
>
> **Question 2 (Misleading CoG Cases)**
> We thank the reviewer for raising the concern that entropy-based CoG selection may overlook critical dimensions. Indeed, some task-essential dimensions do not exhibit low entropy. In Push-2-Box, if a single agent can move a box, box position is important but becomes high-entropy due to frequent solo pushes. In SMAC, enemy positions vary substantially as agents move; in GRF, ball position is naturally high-entropy due to continuous interaction. High entropy in these cases indicates that the dimension is already easy to influence, and empirically we find that excluding such dimensions from CoG does not harm performance.
>
> Once again, we thank the reviewer for the detailed feedback and insightful suggestions, which significantly improved the clarity of the paper.

---

### Official Review · Reviewer_BstB · 2025-10-31

**Soundness:** 2
**Presentation:** 2
**Contribution:** 2
**Rating:** 2
**Confidence:** 3

**Summary:**

The authors propose a intrinsic-based reward shaping method to solve MARL problems with sparse reward. In particular, the intrinsic reward is calculated according to a normalized state difference in each state dimension. States with low entropy will be taken extra care and to be influenced by the agents. Extra pieces are included, e.g., counterfactual rewards, and the method is benchmarked on various simulation environments. Simulations result show that the propose method outperform its baselines.

**Strengths:**

The idea of CoG State Dimension Selection is interesting. The motivation of using intrinsic reward is clear. The paper is easy to follow and derivations and method development are sound.

**Weaknesses:**

I am not fully convinced by the choice/design of CoG State Dimension Selection. In the experiments, the proposed method incurs much larger variance than baselines. The delta definition, i..e, eq(3), is behavior policy dependent. I am not clear how the behavior policy can be systematically selected for various tasks. As the delta value is the key of the proposed method, the authors need to conduct more ablation study of the choice of beta.

**Questions:**

In practice, what are the scales of the H(d) for different dimension? Will the set CoG change a lot during training? How does the choice of beta affect the set CoG, e.g.., more deterministic behavior policy vs random policy?

---

> ### Author Response · Authors · 2025-11-20
> **Response to Reviewer BstB**
>
> Thank you very much for your thoughtful and constructive feedback. We carefully revised the manuscript and prepared the following responses in detail.
>
> **Rationale of CoG dimension selection**
>
> We appreciate the question about why our method focuses on state dimensions that are difficult to influence under the current behavior policy. In the revised manuscript, Section 4.2 includes additional clarification. As widely recognized in Q-learning, visiting a sufficiently diverse set of states is crucial for convergence and avoiding misleading value estimates [R.4]. To encourage such diversity, we use an entropy-based measure that captures how much each state dimension varies under the current behavior policy. In environments like Push-2-Box, dimensions with low entropy identify regions where the policy induces insufficient exploration, and CoG is designed to prioritize these low-entropy dimensions, promoting more diverse transitions and broader state-space coverage. To support this intuition, we added Theorem 4.1, showing that CoG selection is approximately equivalent to choosing the dimension with the smallest normalized next-state entropy, which provides a principled justification for our CoG selection rule.
>
> [R.4] Richard S. Sutton and Andrew G. Barto. Reinforcement Learning: An Introduction. MIT Press, 2018.
>
> **Variance of FIM**
>
> We thank the reviewer for raising this concern on the variance. After re-examining learning curves, we find that, except in environments where baselines fail entirely (e.g., 3s_vs_5z, 27m_vs_30m), FIM’s variance is generally comparable to that of other methods. Where baselines remain near zero performance, their artificially small variance does not indicate stability. In MMM2, FIM shows slightly wider spread, but its entire performance band remains above those of LAIES and other baselines. We therefore believe that the observed variance does not undermine the validity of our conclusions.
>
> **Choice of $\beta$**
>
> We appreciate the inquiry regarding the dependence of $\Delta^d$ on the behavior policy. Our definition intentionally incorporates $\beta$ because CoG aims to measure how difficult it is to influence each dimension under the policy, currently being executed. We do not design or tune a special behavior policy. $\beta$ is simply the $\epsilon$-greedy exploration policy used during training, which transitions from near-random to more deterministic as $\epsilon$ decays. This choice ensures that CoG reflects the agent’s practical controllability at each stage of learning, and we hope this clarification resolves the reviewer’s concern regarding the role of $\beta$.
>
> **Additional ablation for $\beta$**
>
> Following the reviewer’s suggestion, we added dynamic CoG-update experiments in Appendix I.3, where $\Delta^d$ is periodically recomputed under the evolving $\epsilon$-greedy policy. Although this setup recomputes $\Delta^d$ under an increasingly deterministic learned policy, in standard scenario such as 3s_vs_5z, the resulting CoG updates are modest, and the overall learning behavior remains similar to the fixed-CoG version. For simplicity, we therefore keep $\beta$ fixed at its initial form in the main experiments.
>
> **Scale of $\mathcal{H}(d)$**
>
> We thank the reviewer for the question about the practical scale of $\mathcal{H}(d)$. Representative values across dimensions are provided in Fig. 11–12 and Appendix F.3, normalized to the range [0,1] for ease of comparison. In SMAC, frequently varying dimensions such as positions appear with higher entropy, while features that change rarely or only under deliberate coordination (e.g., health) tend to produce lower entropy. Similar distributions arise in other environments in the same appendix, and we hope these visualizations clarify how $\mathcal{H}(d)$ reflects structural variability in each setting.
>
> **Changes in CoG-dimension**
>
> We appreciate the inquiry regarding stability of the CoG set. As detailed in the dynamic CoG analysis in standard 3s_vs_5z (Appendix I.3), the selected dimensions, periodically recomputed under the evolving behavior policy, remain largely stable throughout training. Only a small number of additional dimensions, whose entropies are still relatively high, are incorporated over time. Because intrinsic rewards are modulated by entropy weights as Eq. (9), these adjustments have only limited influence on overall performance, indicating that the CoG set is generally robust throughout training. In the modified 3s_vs_5z setting, where enemy health is initially uninfluenceable and thus absent from the initial CoG, the dynamic CoG update gradually incorporates these newly controllable dimensions and enables the task to be solved.
>
> Once again, we sincerely thank the reviewer for the insightful comments, which significantly improved the clarity and rigor of the paper.

---

### Official Review · Reviewer_r61p · 2025-11-01

**Soundness:** 2
**Presentation:** 1
**Contribution:** 2
**Rating:** 2
**Confidence:** 5

**Summary:**

This paper addresses scenarios in which agents must influence specific state dimensions to accomplish tasks. When these critical dimensions exhibit little variation under normal behavior, agents may fail to discover important state transitions and become trapped in local optima. To tackle this problem, the authors propose the Focusing Influence Mechanism (FIM), which explicitly selects Centers of Gravity (CoG) state dimensions and leverages eligibility traces to enable agents to actively change these otherwise stagnant dimensions, thereby improving exploration efficiency. Specifically, FIM introduces three modules: a state-level focusing mechanism which detects and identifies CoG state dimensions, counterfactual intrinsic rewards which measure each agent's marginal contribution to influencing CoG dimensions, and agent-level focusing mechanism which maintains coordinated and sustained influence via eligibility traces. Experimental results across several multi-agent reinforcement learning benchmarks show that FIM can achieve more efficient cooperative performance compared to existing methods.

**Strengths:**

The paper conducts experiments across multiple scenarios and demonstrates improved empirical performance, which supports the practical relevance of the approach.

**Weaknesses:**

1. **Motivation example is not fully convincing**

    The Push-2-Box example used to motivate the work is a highly contrived setting. In this environment, the design of the intrinsic rewards indirectly informs agents about the task goal, leading to strong performance. As a result, the observed success does not convincingly validate the general usefulness of CoG-based design (see Questions 3–6 below).

2. **Intrinsic rewards are task-specific and may encode task goals implicitly**

    In Push-2-Box example, the x and y positions of the boxes are chosen as CoG state dimensions, and the intrinsic rewards are directly designed to encourage agents to modify these values. Naturally, this leads to successful task completion. However, this is essentially an implicit encoding of the task objective, which could even be considered “cheating.” One could directly define intrinsic rewards proportional to the distance between the box and its original location, achieving similar results. Consequently, this example does not convincingly demonstrate that CoG-based design inherently improves task performance. I recommend using examples in the Multi-Agent Particle Environment to illustrate the motivation.

3. **Notation is confusing**

    The preliminary section should clearly define state dimensions and associated symbols (e.g., $D$), since the expression $s_t = (s^0_t, s^1_t, \dots, s^{D-1}_t)$ is ambiguous, and $s^i_t$ is not defined. Similarly, the usage of $\beta$ lacks clear explanation (see Question 2  below).

**Questions:**

1. In Figure 1, if a single agent can push the box (although slower), why does vanilla QMIX fail to move the box? Theoretically, one agent pushing one step at a time versus two agents pushing simultaneously should require similar effort. Why does vanilla QMIX not achieve the performance shown in Figures 1(c) and 1(d)? The motivation and the core reason why focusing on specific state dimensions improves performance remain unclear.
2. In Equation 3, is $\beta$ an individual agent’s policy or the joint policy of all agents? My understanding is that it represents the joint policy, which seems inconsistent with the definitions in the preliminaries.
3. Do the state dimensions have concrete meaning after encoding? Why does keeping ${CoG}_\delta$ fixed during training still yield strong performance? This makes it difficult to argue that the experimental results are due to CoG discovery.
4. On line 266, the authors state that agents frequently switch between two boxes and fail to push either to the wall due to multiple CoG state dimensions, motivating the AFI mechanism. Can this mechanism be considered general, or is it only effective for a small class of scenarios? For instance, if there are three agents and three boxes, and pushing the boxes fastest only requires two agents, the method may waste time and even perform worse than each agent pushing one box.
5. In the box pushing example, what is the difference between two agents jointly pushing a box versus each agent pushing a separate box? Why must two agents push together? The main challenge is for agents to understand the task goal; once understood, the two setups montioned above are effectively equivalent. The apparent effectiveness of your method in the box-pushing example arises because the intrinsic reward directly encodes the task goal.
6. The method focuses on encouraging agents to change dimensions that are difficult to influence under the current policy. What is the fundamental rationale for doing so? Why should this lead to better task performance? (Please avoid using the box pushing example, since in that case changing those two dimensions directly corresponds to the task.)
7. In SMAC, why are the rewards for winning and losing battles the same (+1 and -1), whereas in GRF they differ by a factor of 100 (+100 and -1)?

---

> ### Author Response · Authors · 2025-11-20
> **Response to Reviewer r61p (1/2)**
>
> We sincerely thank the reviewer for the time and care invested in reading our work and providing detailed, insightful feedback. Your comments have meaningfully improved both the clarity and completeness of the paper.
>
> **Weakness 1, 2/Question 5 (Motivation example)**
>
> We appreciate the concern regarding the motivation example and the design of the Push-2-Box environment. The purpose of the motivation section is to highlight a situation where existing MARL methods struggle, explain why this happens, and motivate our proposed solution. Push-2-Box is used solely as a concrete and interpretable example for this purpose. It is not an ad hoc environment crafted to favor our method, but a straightforward modification of the standard PushBox setup widely used in prior MARL work [R1–R3]. We simply add a second box and adopt a sparse reward scheme. As is common in MARL benchmarks, PushBox-style tasks require nontrivial cooperation: a box moves only when multiple agents push it together. If a single agent could reliably move the box and solve the task, the scenario would not meaningfully test multi-agent cooperation. From this perspective, Push-2-Box is a natural testbed to examine whether agents can learn to coordinate on a shared target under sparse rewards.
>
> We also emphasize that our intrinsic reward is not engineered to directly encode the task goal. It does not, for instance, encourage movement toward the wall or along any goal-aligned direction. Instead, it encourages agents to interact more frequently with state dimensions that are hard to change under random behavior, such as box positions. Whether those interactions ultimately lead to the box reaching the wall is determined by the environment dynamics and the external reward. Thus, the example does not rely on task-specific shaping of the intrinsic signal, but illustrates that FIM improves cooperative exploration in a cooperation-critical setting that extends an existing MARL benchmark. At the same time, we agree that Push-2-Box alone cannot establish the generality of our approach. It is used to build intuition and visualize the effect of each component, while the main empirical validation on SMAC and GRF provides evidence in substantially more complex MARL benchmarks, further supported by our responses to Questions 3, 4, and 6.
>
> [R1] Tonghan Wang, Jianhao Wang, Yi Wu, and Chongjie Zhang. Influence-based multi-agent exploration. ICLR, 2020.
> [R2]. Lou-Jen Liu, Unnat Jain, Raymond A. Yeh, and Alexander Schwing. Cooperative exploration for multi-agent deep reinforcement learning. ICML, 2021.
> [R3] Pei Xu, Junge Zhang, Qiyue Yin, Chao Yu, Yaodong Yang, and Kaiqi Huang. Subspace-aware exploration for sparse-reward multi-agent tasks. AAAI, 2023.
>
> **Question 3 (CoG state dimensions)**
>
> We appreciate the concern regarding the meaning and stability of state dimensions. In our settings (Push-2-Box, SMAC, and GRF), the state representation is based on hand-crafted features provided by the environments (positions, health, shields, etc.), so each dimension has a clear semantic interpretation. As mentioned in Section 4.2, we do not assume that CoG state dimensions must remain fixed in general, and our framework conceptually allows CoG to be updated dynamically if the task or policy induces significant shifts in which dimensions are critical. We also tested an online CoG update scheme in Appendix I.3 and found that, for the environments considered in this paper, dynamically updating CoG dimensions brought little additional benefit compared to using a fixed set. For this reason, we chose the fixed-CoG variant in our main experiments, primarily to keep the implementation simple, rather than because the method inherently relies on CoG being fixed. Still, in the more complex task that we designed, in which the target of influence must be changed, dynamic CoG update gradually incorporates newly controllable dimensions and enables the task to be solved, as shown in same appendix.
>
> **Question 1 (Failure of QMIX in Push-2-Box)**
>
> We appreciate the question regarding why vanilla QMIX fails to move the box in Figure 1. As discussed in Section 4.1, solving Push-2-Box requires both agents to consistently target and push the same box over a long horizon before any reward is observed. Although a single agent can slightly move the box, it cannot move it a meaningful distance unless the other agent pushes the same box simultaneously, and the wall is far enough that an episode where one agent alone accidentally pushes the box all the way is extremely rare, as reflected in the visitation counts in Figure 1. Consequently, QMIX almost never receives informative rewards in this sparse-reward setting and fails to learn the required cooperative behavior, whereas our method explicitly encourages coordinated pushing around the box and thus enables successful completion of the task, illustrating the importance of cooperation in MARL.

---

> ### Author Response · Authors · 2025-11-20
> **Response to Reviewer r61p (2/2)**
>
> **Question 4 (Generality of AFI)**
>
> Thank you for raising this point about the generality of AFI beyond the Push-2-Box setting. First, AFI does not rigidly force all agents to focus on a single state dimension at all times. Rather, it biases exploration so that agents are more likely to jointly interact with CoG state dimensions that are estimated to be important. In the reviewer’s example with three agents and one box, AFI does not require all three agents to always push the same box. Instead, it increases the chance that multiple agents try to influence a high-priority CoG dimension together. If, in the course of learning, the best solution is that two agents push the box while the third takes a different action that yields higher extrinsic reward, the underlying MARL algorithm is free to adopt that behavior, and AFI does not prevent this.
>
> Our motivation for AFI is that, under sparse rewards, baselines such as QMIX rarely discover and repeatedly visit such coordinated configurations, simply because the probability of reaching them through unguided exploration is very low. AFI makes these coordinated patterns easier to discover and stabilize, which we believe is a general benefit rather than something tied only to Push-2-Box. As discussed in Section 5.3, in more complex environments such as SMAC, AFI does not collapse all agents onto a single target in a trivial way. Instead, in some scenarios agents learn behaviors where part of the team lures or positions while others focus fire, indicating that AFI can support rich cooperative strategies rather than enforcing a single rigid pattern.
>
> **Question 6 (Rationale of CoG dimension selection)**
>
> We appreciate the reviewer’s question regarding why our method focuses on state dimensions that are difficult to influence under the current behavior policy. In the revised version, we have added clarifications to Section 4.2 (CoG) to address this point. In Q-learning based methods, it is important to visit as diverse a set of states as possible in order to improve convergence [R.4]. To encourage such diverse state visitation, we consider an entropy based approach that captures the information content of state transitions. As illustrated in challenging examples such as Push-2-Box, given a fixed behavior policy, if we measure the change of each state dimension under this policy, those dimensions whose entropy is low relative to their average magnitude of change correspond to regions where the induced state transitions are not sufficiently diverse. Our CoG mechanism is therefore designed to focus on actively exploring and diversifying these low-entropy dimensions, so that the agent can reach a wider variety of states more efficiently. To theoretically support this intuition, we added Theorem 4.1, which shows that the proposed CoG state-dimension selection is approximately equivalent, for a given state and behavior policy $\beta$, to choosing the dimension whose next-state distribution has the smallest normalized entropy. This result provides a theoretical motivation for our CoG based selection rule, which is designed to encourage more effective state-space exploration.
>
> [R.4] Richard S. Sutton and Andrew G. Barto. Reinforcement Learning: An Introduction. MIT Press,
> second edition, 2018.
>
> **Weakness 3/Question 2 (Notation clarification)**
>
> We thank the reviewer for pointing out the ambiguity in our notation. In the revised manuscript, we now explicitly define $D$ as the total number of state dimensions, and clarify that $\boldsymbol{\beta}$ denotes the joint behavior policy used to sample transitions for estimating state-change entropy. We appreciate the reviewer’s comment, which helped us improve the clarity of our notation.
>
> **Question 7 (Reward design of SMAC/GRF)**
>
> We thank the reviewer for this question and appreciate the opportunity to clarify the reward design used in our experiments. For both SMAC and GRF, we follow the reward settings from the prior LAIES work. In SMAC, the outcome of combat is relatively close to deterministic, so a symmetric $\pm 1$ reward is sufficient. In contrast, GRF is more stochastic and challenging, so the previous work uses a larger positive reward (for example, $+100$ for scoring) to provide a stronger learning signal, and we adopt the same design. We hope this clarification addresses the reviewer’s concern.
>
>
> Once again, we are grateful for the reviewer’s thoughtful comments and constructive suggestions, which greatly strengthened the final version of the manuscript.

---

### Author Response · Authors · 2025-11-20
**General Response to All Reviewers**

We sincerely thank all reviewers for their constructive feedback. Following your suggestions, we have substantially strengthened the manuscript with additional descriptions to clarify the rationale of our framework and further experiments to improve its empirical support and overall clarity. A revised version, with all changes highlighted in blue, has been uploaded. The major updates are summarized below.

**(i) Rationale of CoG dimension selection (Section 4.2, Appendix C):** We strengthened the theoretical rationale for CoG from an information theoretic perspective, emphasizing the need for sufficient exploration to ensure good convergence in value based methods. We also clarified how our CoG criterion, defined via the entropy of dimension wise state differences, contributes to increasing the entropy of future states and explained why normalization of these entropy values is necessary for stable and comparable dimension selection.

**(ii) Dynamic CoG dimension selection (Section 4.2, Appendix I.3):** We explicitly clarified that CoG state dimensions need not be fixed and that a dynamic CoG variant can be considered within our framework. We compared a fixed CoG set chosen under the initial policy with a periodically updated CoG set during training and found little difference on standard MARL benchmarks, which justifies our use of the fixed setup in the main experiments. We also presented a scenario where dynamic CoG is advantageous and empirically compared the two variants in that setting.

**(iii) Additional experiments and analyses (Section 5, Appendix D.1, I):** To further isolate and support the roles of SFI and AFI, we extended the component analysis by injecting our SFI and AFI modules into the prior method LAIES and directly comparing the variants. These results more firmly demonstrate that both components are important for performance. In addition, we incorporated several other experiments suggested by the reviewers, which together provide a clearer and more comprehensive empirical picture of our framework.

We believe these revisions address the main concerns raised during review and improve the clarity and completeness of the paper. We are grateful for the reviewers’ guidance, which materially enhanced the manuscript.

---

### Author Response · Authors · 2025-12-02

Dear Area Chair,

Thank you very much for your time and effort in handling our submission. In addition to the reviewer-specific replies and common responses provided below, we first briefly summarize the main contribution of our work and how we addressed the reviewers’ concerns, so that the revisions are easier to follow from your perspective as Area Chair.
***
**Main contribution:** In long-horizon sparse-reward multi-agent environments, we observe that standard MARL policies often fail because they do not sufficiently explore certain state dimensions that are particularly difficult to change. To address this issue, **the proposed FIM framework identifies, for a given policy, low-entropy state dimensions that are hard to vary and defines them as Center-of-Gravity (CoG) dimensions. FIM then introduces two focusing influence components, SFI and AFI, that enable agents to concentrate their influence on these CoG dimensions**. SFI designs a counterfactual reward that encourages each agent to focus its influence on the CoG dimensions, while AFI builds on eligibility traces to guide multiple agents to jointly concentrate their influence on these dimensions, thereby improving coordinated exploration. As a result, unlike existing baselines that either focus on irrelevant parts of the state space or fail to coordinate on the critical dimensions, FIM explicitly targets underexplored dimensions in difficult tasks and drives agents to change them, leading to substantially improved performance on challenging sparse-reward MARL benchmarks such as Push2Box, sparse SMAC, and GRF, and providing a meaningful contribution to the MARL domain.
***
**Summary of rebuttal:** From the reviews, our understanding is that most reviewers agreed that FIM provides an original contribution with strong empirical performance. The remaining concerns were mainly about the formal justification for CoG selection (reviewers *r61p*, *BstB*, *XY1P*), the analysis of dynamically updating CoG (reviewers *r61p*, *BstB*, *Mvru*), and the need for additional clarification and ablations (reviewers *r61p*, *XY1P*, *Mvru*). In response, we introduced Theorem 4.1, which provides **an entropy-based upper bound argument showing that CoG dimensions correspond to underexplored state dimensions** for the current policy and clarifies the need for normalization across dimensions for fair comparison. We also added **experiments that periodically update the CoG state dimensions and compare them with the fixed-CoG setup to reveal when dynamic updates are beneficial**, clarified why the motivation example is a representative and important case for our method, and applied the FIM components SFI and AFI to the existing LAIES baseline to compare their individual benefits. We believe these additions substantially improve the clarity and rationality of the proposed method.
***
Due to an OpenReview system issue, it is unfortunate that we could not receive follow-up comments from any of the reviewers, and the reviews are currently somewhat split. The most substantial concerns were raised by reviewers *r61p* and *BstB*, who focused on the rationality of CoG dimension selection and the effect of dynamic updates, and **we expect that the added theorem and new dynamic-update experiments largely address these points**. Reviewer *r61p* also expressed a concern that Push2Box might be an environment specifically constructed for our method; in the rebuttal, we clarified that it is a minor modification of an existing environment and that **our entropy-based approach is intended to be broadly applicable rather than tailored to this particular case**, and we hope this helps alleviate that concern. For reviewers *XY1P* and *Mvru*, who already had relatively positive impressions, the remaining issues beyond these common points were mainly requests for additional experiments, which we have carried out and analyzed in detail, so we believe their concerns would also be satisfactorily resolved and their positive views maintained.

In summary, we believe that FIM offers a clear and distinctive contribution, and that the rebuttal process has allowed us to substantially address the central theoretical and dynamic-update concerns, thereby further strengthening the paper and clarifying its main messages in light of the reviewers’ feedback. We hope this overview is helpful for your decision, and we are sincerely grateful for your time and careful consideration of our work.

---

### Meta-Review · Area_Chair_GhGv · 2025-12-19

**Summary:**

This submission presents the Focusing Influence Mechanism (FIM), a novel framework designed to address the core challenge of cooperative multi-agent reinforcement learning (MARL) under sparse reward settings—namely, inefficient exploration and inadequate coordinated attention among agents.
In response, the authors added Theorem 4.1 to provide information-theoretic support for CoG selection, conducted dynamic CoG update experiments, etc.. These revisions effectively addressed many technical and empirical concerns raised by reviewers. However, Several areas warrant further refinement: first, the formal theoretical link between low-entropy dimensions and task-criticality would benefit from additional rigorous validation; second, supplementary quantitative evidence would strengthen the case for hyperparameter transferability across novel environments. By further strengthening the theoretical grounding, etc., this research is well-positioned to deliver substantial academic impact in future submissions.

**Reviewer Concerns:**

1. Extreme Initial Policy Bias
2. Hyperparameter Transferability Quantification

**Reviewer Scores:**

r61p: 2.
BstB: 2.
XY1P:6.
Mvru:6.

---

### Decision · Program_Chairs · 2026-01-26

Reject